# Extended Evaluation of Viral Diversity in Lake Baikal through Metagenomics

**DOI:** 10.3390/microorganisms9040760

**Published:** 2021-04-05

**Authors:** Tatyana V. Butina, Yurij S. Bukin, Ivan S. Petrushin, Alexey E. Tupikin, Marsel R. Kabilov, Sergey I. Belikov

**Affiliations:** 1Limnological Institute, Siberian Branch of the Russian Academy of Sciences, Ulan-Batorskaya Str., 3, 664033 Irkutsk, Russia; ivan.kiel@gmail.com (I.S.P.); sergeibelikov47@gmail.com (S.I.B.); 2Institute of Chemical Biology and Fundamental Medicine, Siberian Branch of the Russian Academy of Sciences, Lavrentiev Ave., 8, 630090 Novosibirsk, Russia; alenare@niboch.nsc.ru (A.E.T.); kabilov@niboch.nsc.ru (M.R.K.)

**Keywords:** viral diversity, metagenomic analysis, freshwater viromes, lake ecosystems, Lake Baikal

## Abstract

Lake Baikal is a unique oligotrophic freshwater lake with unusually cold conditions and amazing biological diversity. Studies of the lake’s viral communities have begun recently, and their full diversity is not elucidated yet. Here, we performed DNA viral metagenomic analysis on integral samples from four different deep-water and shallow stations of the southern and central basins of the lake. There was a strict distinction of viral communities in areas with different environmental conditions. Comparative analysis with other freshwater lakes revealed the highest similarity of Baikal viromes with those of the Asian lakes Soyang and Biwa. Analysis of new data, together with previously published data allowed us to get a deeper insight into the diversity and functional potential of Baikal viruses; however, the true diversity of Baikal viruses in the lake ecosystem remains still unknown. The new metaviromic data will be useful for future studies of viral composition, distribution, and the dynamics associated with global climatic and anthropogenic impacts on this ecosystem.

## 1. Introduction

Viruses (mainly bacteriophages) are the most numerous and diverse components of aquatic ecosystems. They significantly affect the biodiversity, composition, and productivity of aquatic ecosystems [1,2,3,4].

The rapid progress in aquatic virology several decades ago has become possible thanks to the development of methods and devices for virus counting, filtering, and concentration of virus-like particles (VLPs), and most importantly, to new genomic technologies, including high-throughput sequencing and new approaches to library construction [5,6]. Metagenomics, i.e., sequencing of the genomic content of an uncultured community, now represents a powerful approach for investigating the diversity and relationships of viruses in aquatic ecosystems.

Since the first metavirome reports [7,8], numerous studies have been carried out in various biomes; each of them has provided new information about the diversity and roles of viruses in natural ecosystems. Marine studies encompass a wide variety of seas and oceans around the world, with a large number of dedicated stations and samples from different depths, seasons, etc. [6]. In contrast to marine viruses, freshwater viruses remain less studied; however, the interest in freshwater ecosystems, including large water bodies of great social importance, is gradually increasing [9,10,11,12,13,14,15,16,17,18].

Baikal is a unique, ancient, oligotrophic freshwater lake—the world’s deepest, oldest, and largest by volume. The lake is characterized by an unusual climatic environment and amazing biological diversity (mainly endemic flora and fauna) [19,20]. Some parameters (oligotrophy, large depth, and oxygen concentration) and dynamic processes occurring in the lake (horizontal cyclonic currents, stratification, thermal bar, spring and autumn homothermy, upwellings and downwellings, and the deep convection) [21] are similar to oceanic parameters. Lake Baikal has the shape of a giant narrow crescent stretching from southwest to northeast for 636 km. The lake consists of three basins (southern, central, and northern) that are separated by underwater elevations. There are many large and small bays in the water area of Lake Baikal. The distinctive geomorphological, climatic, hydrophysical, and hydrochemical parameters are typical of the various areas and basins of the lake. Lake Baikal freezes in the first half of January (ice thickness up to 130 cm); the ice breaks in early May. During the warm period (July to September) the temperature of the surface-water layer in open Baikal is 5 to 19 °C, and the temperature of the deep-water layer is circa 4 °C. [20].

The first viral genetic studies showed a high phylogenetic diversity of T4-like bacteriophages and cyanophages from the Myoviridae family in the pelagic and coastal zones of Lake Baikal based on *g23* and *g20* marker-gene sequencing [22,23]. The phage gene assemblages are different in the lake basins; the environmental factors specifying the trophic status of the lakes influence the diversity of viral communities and may be of greater importance than geographical patterns [24]. Previous studies [25,26] demonstrated through the UniFrac method that *g23* and *g20* assemblages from freshwater lakes, including Baikal ones, were more closely related to those from terrestrial aquatic environments (wetlands, paddy fields, and upland soils) than to those from marine ones.

The first metagenomic study of the Lake Baikal virome was carried out in 2013 [27]. We investigated the sample of coastal water from the southern basin of the lake and revealed a large potential diversity of viruses of different families and genera, affecting a wide range of bacteria, archaea, algae, amoeba, flagellates, fish, crustaceans, insects, etc. [27]. Recently, Potapov et al. [28] examined two more viromes from the photic layer of the pelagic zone of the lake during the under-ice and late-spring periods and revealed some differences in the taxonomic and functional composition of viruses in these datasets. A comparative analysis of viral communities from different environments indicated different distribution patterns in soil, marine, and freshwaters. Baikal viromes formed a separate clade with those from the Great Lakes of North America (Michigan, Erie, and Ontario).

The aim of this study was to more broadly estimate the viral diversity in Lake Baikal and compare viral communities (mainly dsDNA viruses) in different areas of the lake. We assumed and confirmed that the wide variety of habitats and conditions within the Baikal ecosystem determines the formation of diverse and distinct viral communities. We also expected to identify the special compositions of Baikal viruses, due to the extreme environment and high levels of endemism in the lake. The low level of similarity of the detected viral reads and scaffolds with known viral genomes partially confirmed the uniqueness of the viral populations. However, the chosen approach based on comparing viromes with known viral genomes did not allow us to fully verify this hypothesis, and this issue requires additional research.

## 2. Materials and Methods

### 2.1. Description of Study Sites and Sample Processing

Water samples were collected at deep-water and shallow stations of Lake Baikal in September 2014. Deep-water stations were located in the southern (central site of the Listvyanka–Tankhoy section and Listvennichny Bay) and central (the Ukhan–Tonky section) basins of the lake (Figure 1, Table 1). The southern and central basins are the deepest, 1423 m and 1637 m, respectively [20]. Listvennichny Bay is one of the largest (with a depth over 1000 m); it is located near the Angara River source (Figure 1). Two shallow stations (Olkhonskiye Vorota Strait and Kurkut Bay) were located in the area of the Maloye More Strait (Figure 1, Table 1). The Maloye More Strait is one of the largest shallow areas (maximum depth 200 m) located between the west coast of the lake and the largest Olkhon island. The Olkhonskiye Vorota Strait (depth down to 100 m) [20] connects the Maloye More Strait with the central basin of Lake Baikal (Figure 1). There are many sunny days in this region; in summer, the water in numerous bays warms up to +20–24 °C.

At deep-water stations, the sampling was carried out from the board of the research vessel “G.Yu. Vereshchagin” using an SBE 32 Carousel Water Sampler bathometer system (USA) at depths of 0, 5, 10, 15, 25, 50, 100, 250, and 500 m; 2.5 liters of water were taken from each depth at these stations. The water samples were filtered to remove bacteria and other organisms larger than viruses through 0.45 µm nitrocellulose membrane filters (47 mm diameter, Vladisart, Vladimir, Russia) using a vacuum pump. Then, the water samples from different depths of each station were combined in a 25 l glass bottle, separately for each station (samples V1, V2, and V4; Table 1), and VLPs were concentrated in the resulting integrated samples (the total volume of each sample was approximately 22 l) with the Sartocon Slice tangential filtration pilot system (30 kDa, Sartorius, Germany) to 100 mL, filtered through the 0.2 µm syringe filters (PES, Sartorius) and additionally concentrated with the Amicon-15 ultrafiltration centrifugal devices, 30 kDa (Millipore, Darmstadt, Germany) to 1.5 mL as previously [27]. At shallow stations, the sampling was carried out using a five-liter bathometer and a winch. The sampling depths at shallow stations were as follows: 0, 5, 10, 15, 25 m in the Olkhonskiye Vorota Strait (three liters from each depth), and 0 m at the central site of Kurkut Bay. The prefiltered water samples (through 0.45 µm nitrocellulose membrane filters) from different depths at these two sites were combined (sample V3; Table 1), and the integrated sample (approximately 18 l) were processed as described above for the samples V1, V2, and V4.

Combining samples from different depths allowed us to identify the most complete variety of viral communities from different stations with the least number of samples and resources for their processing and analysis. However, this approach cannot trace the vertical distribution of viral communities and assess the differences in their composition in the surface and deep layers of the lake, which requires additional research.

### 2.2. DNA Extraction

The concentrates of VLPs were treated with DNase I (50 U mL^−1^; Thermo Fisher Scientific, Carlsbad, CA, USA) for 1.5 h to remove contaminating external nucleic acids. DNA was isolated using sodium dodecyl sulfate (SDS), proteinase K, and the phenol-chloroform extraction method [29]. The absence of bacterial contamination in the VLPs concentrate was checked by PCR amplification with the 16S rRNA universal primers, 27F and 1093R, as described in [30]. The concentration and quality of the extracted DNA were measured with a NanoDrop spectrophotometer (Thermo Fisher Scientific, Carlsbad, CA, USA).

### 2.3. Library Preparation and Sequencing

The total viral DNA was sheared in a microTUBE AFA Fiber Snap-cap using a Covaris S2 instrument (Covaris, Woburn, MA, USA) with a medium-size distribution of fragments of approximately 500 bp. The paired-end libraries were prepared using a NEBNext Ultra DNA library prep kit for Illumina (New England Biolabs, Ipswich, MA, USA).

Sequencing of the libraries was performed on a MiSeq genome sequencer using the MiSeq Reagent Kit v.3 (2 × 300 cycles, Illumina) in the Genomics Core Facility (Institute of Chemical Biology and Fundamental Medicine, Siberian Branch of the Russian Academy of Sciences, Novosibirsk, Russia).

Unprocessed virome reads for the samples V1–V4 from Lake Baikal were submitted to the National Center for Biotechnology Information (NCBI), Sequence Read Archive (SRA) database (BioProject PRJNA398439, BioSamples SAMN15770410-SAMN15770413).

### 2.4. Preparation of Reads

The general scheme of bioinformatic data processing is shown in Appendix A.

For comparative analysis, we also used the published datasets on freshwater viromes, including Baikal ones [27,28], sequenced using the same library preparation and sequencing techniques as in our study (the Illumina platform) and available in the NCBI SRA database at the time of our analysis (Table 2).

SRA archives of data on other freshwater bodies (FASTQ files) were downloaded from the NCBI database using the “fastq-dump” utility. All data, including those from Lake Baikal and other lakes, were grouped into one FASTQ dataset for future analysis.

The visualization of the quality of the reads was carried out using the FASTQC program (http://www.bioinformatics.babraham.ac.uk/projects/fastqc; accessed on 8 January 2019). Trimming by the quality of reads was carried out in the Trimmomatic V 0.39 program [31] using the adaptive quality trimmer options (MAXINFO: 40: 0.1); the reads of 100 bp or more were used for further analysis.

### 2.5. Taxonomic Analysis of Viral Reads

For taxonomic identification, the metagenomic reads were compared with the NCBI RefSeq complete viral genome database and the NCBI RefSeq complete viral proteome database [32] (released on 9 February 2020). A double comparison was necessary because some complete viral genomes of NCBI RefSeq were not annotated, and reads similar to these genomes would have been left out of the analysis (as unidentified) if the RefSeq viral proteome database only was used. The comparison was carried out in two stages. At the first stage, each read was compared with the DIAMOND program [33] against the NCBI RefSeq complete viral proteome. For the DIAMOND program, we used the “more-sensitive” option. Then, we selected reads with proteomic similarity ≥ 35%. If there was no similarity of a read with any on the NCBI RefSeq complete viral proteome database, such read passed to the second stage. At the second stage, the read was compared with the NCBI RefSeq complete viral genome database by the BLASTn algorithm [34]. The BLASTn parameters used were as follows: cost to open a gap, 2; gap extension cost, 1; word size, 9; penalty for a nucleotide mismatch, 1; reward, 1; e-value of ≤ 0.00001, bit score ≥ 50. If no similarity with the database was found at this stage, the read was not identified as viral. The reads similar to the same viral genome (ID) from the NCBI RefSeq database, were summarized and assigned to one virotype (virus species from the NCBI RefSeq database). The count of the virotypes in a sample was defined as the number of reads in the sample associated with this virotype by the DIAMOND or BLASTn program. Consequently, the count table of the various virotypes in the samples was obtained.

The count table of virotypes (number of hits per each virotype in the sample) was normalized to the genome length of the virus, representing the virotypes in the RefSeq database. Normalization was carried out according to the following algorithm:

(1)The average length of virotype genomes was calculated by Formula 1:(1)$$Ls=\frac{1}{N}\overset{N}{\underset{i=1}{\sum }}L_{i}$$
where Ls is the average length of genomes in a sample, N—number of virotypes in the sample, $L_{i}—$genome length of the i-th virotype.


(2)The normalization coefficient $S_{i}$ for each virotype in the sample was calculated by Formula 2:
(2)$$S_{i}=\frac{Ls}{L_{i}}$$


(3)Count of each virotype in the sample was recalculated using Formula 3:(3)$$nk_{i}=\frac{n_{i}}{S_{i}}$$
where $n_{i}$ is the initial value of the virotype count in the sample; $nk_{i}$—the virotype count in the sample normalized to genome length.

Normalization to the genome length was used because viruses with longer genomes obviously make a greater contribution to the DNA pool at the same concentration of viral particles in the sample. A similar RPKM normalization is used in the transcriptomic analysis.

The algorithm of taxonomic identification used in our study was similar to that of the known METAVIR 2 service [35], which identifies a read based on comparison with the complete viral proteomes. The advantage of our algorithm is the combined analysis of complete viral genomes and proteomes that allowed us to compare the reads with viral genomes, for which the annotation of their proteome was not represented in the NCBI RefSeq database, and, at the same time, to identify the distant similarity of the reads, comparing translated reads with proteins. Moreover, the scheme used allowed us to apply available computing resources with a large number of computing cores and a small amount of RAM.

For further statistical calculations, we took into account only 95% of virotypes, mostly represented by the number of reads.

The pipeline for processing the BLASTn and DIAMOND results were created using the R programming language (R Project for Statistical Computing: https://www.r-project.org/; accessed on 1 June 2020) [36].

### 2.6. Statistical Analysis of Taxonomic Diversity

A virotype count table was used for comparative statistical analysis of viral community samples. We evaluated the potential (underestimated) number of virotypes (α-diversity or species richness) in communities, with the increasing reading depth using Chao1 [37]. Shannon and Simpson indices [38] of biodiversity were also calculated for each sample.

For multivariate statistical analyses, the taxonomic composition based on the virotype count table was normalized to the relative abundance of reads per sample. To equalize the effect of virotypes with different counts per sample (from the highest to the lowest ones), the values ranged between 0 and 1.

The taxonomic composition similarity of the samples was visualized using the non-metric multidimensional scaling (NMDS) ordination method with the Euclidian distance metric. Gradient vectors of the viral family composition and community biodiversity indices were fitted on the NMDS scatter plot. The reliability of linear approximation for gradient vectors was assessed by multivariate linear regression analysis.

Biodiversity analysis and NMDS were carried out in the “vegan” package for the R programming language [39], according to the tutorials [40].

The samples were clustered by similarity counts of virotypes using the Euclidian distance metric and “average” hierarchical cluster method by standard functions of the R programming language.

Dominant virotypes from the virotype count table in each sample were visualized with the heat map generated using the “gplots” [41] package in R. Rows (virotypes) and columns (samples) were clustered and grouped in similarity order (i.e., Euclidean distance metric and the complete-link clustering method).

The significance of the difference between the samples in counts of virotype reads was assessed using the chi-square test for independence. The *p*-value for the chi-square test was adjusted by the Bonferroni correction for multiple hypothesis testing.

### 2.7. De Novo Assembly of Metagenomic Reads and Identification of Viral Scaffolds

The SPAdes 3.13.1 metagenomics assembler, metaSPAdes [42], with parameters of paired-end reads and K-mer lengths of 21, 33, 55, and 77 were used for the de novo cross-assembly of Baikal samples.

The Burrows-Wheeler Aligner (BWA) software [43] was used to map paired-end reads on scaffolds and calculate the total coverage of scaffolds in the assembly and coverage of scaffolds by reads from each sample. The scaffolds with total coverage of more than five and a length of ≥5000 nucleotides were used for analysis.

We identified the viral scaffolds and open reading frames (ORFs) in them using the VirSorter tool [44] on the «CYVERSE» Discovery Environment webserver (https://de.cyverse.org/de/; accessed on 19 July 2020).

The BWA results were used to determine the number of reads mapped on each predicted viral scaffold from each sample. Counts of the predicted viral proteins (ORFs) in samples were defined as the number of reads mapped on a scaffold containing a given protein. Consequently, the count table of viral scaffold representation in the analyzed samples was constructed.

### 2.8. Taxonomic Identification of Viral Scaffolds

Taxonomic identification for the viral scaffolds represented in the Baikal samples was carried out by comparisons of predicted viral proteins in scaffolds with the NCBI RefSeq complete viral proteome database. The comparison was carried out by the BLASTp algorithm with the following parameters: word size, 6; gap open cost, 6; gap extension cost, 2; e-value ≤ 0.00001; bit score ≥ 50, and identity ≥ 35%. For each protein in the scaffold, the best match in terms of the bit score value was selected. If a single scaffold had multiple proteins that matched different taxa (NCBI RefSeq ID), the one with the largest number of matching proteins was chosen as the most closely related virus taxon (virotype) of this scaffold. If the proteins were not repeated in the match list, the level of similarity of the matched proteins was taken into account, and the NCBI RefSeq taxon (ID) with the highest percentage of protein similarity was selected as the virotype.

### 2.9. Functional Annotations of the Viral Communities

The predicted viral proteins were matched with the UniProtKB/Swiss-Prot database [45] by the BLASTp algorithm with the following parameters: word size for word finder algorithm, 6; cost to open a gap, 6; gap extension cost, 2. Viral proteins were considered identified if the best hits had e-value ≤ 0.00001, bit score ≥ 50, and identity ≥ 35%. KEGG (Kyoto Encyclopedia of Genes and Genomes) Orthology (KO) protein identifiers were taken from annotations uploaded from UniProtKB/Swiss-Prot [46], and processed in the «KEGGREST» package [47] for R programming language to obtain the KEGG pathway classification (https://www.genome.jp/kegg/pathway.html; accessed on 4 July 2020) of the predicted viral proteins. The count of the predicted viral proteins in samples was transformed in counts of the KEGG pathway classification groups that were normalized for the average number of hits on the viral proteins in each sample.

The predicted viral proteins were matched with the clusters of orthologous groups—COG [48] database by the BLASTp algorithm. For BLASTp, we used the following parameters: word size for word finder algorithm, 6; cost to open a gap, 6; gap extension cost, 2. Viral proteins were considered identified if the best hits had e-value ≤ 0.00001, bit score ≥ 50, and identity ≥ 35%. Counts of predicted viral proteins in each sample were used to calculate the general functional COG categories. Counts of COG protein categories in each sample were normalized for the average number of hits on the predicted viral proteins.

KEGG pathway classification groups and counts of the COG protein category in each sample were visualized with a heat map generated using the «gplots» package [41] in R. Rows (protein functional categories) and columns (samples) in the protein classification count tables were clustered and grouped in similarity order (i.e., Euclidean distance metric and the complete-link clustering method).

VIBRANT v. 1.2.1 software (https://github.com/AnantharamanLab/VIBRANT; accessed on 2 March 2021) [49] was used to determine auxiliary metabolic genes (AMGs) among predicted viral proteins that affect the metabolic activity of the host community. VIBRANT was run with a single thread and all settings set to default as recommended in the manual. AMG functional gene categories were classified according to the KEGG functional identifier. The count of the AMG viral proteins in samples was transformed in counts of the KEGG pathway classification groups that were normalized for the average number of hits on the viral proteins in each sample.

### 2.10. Host Prediction

Host prediction for Baikal viral scaffolds was carried out by two methods. The first method was based on the Virus–Host database [50]. After taxonomic identification of predicted viral scaffolds (see section “Taxonomic identification of viral scaffolds”), the spectrum of corresponding hosts from the Virus–Host database was obtained. The second method was based on the VirHostMatcher-Net software [51] with default settings in “short contig” mode. Data preparation and running were performed according to the manual (https://github.com/WeiliWw/VirHostMatcher-Net; accessed on 4 March 2021). This software provides a prediction of the prokaryotic host based on 62,493 genomes of bacteria and archaea with a series of models: CRISPR (Clustered Regularly Interspaced Short Palindromic Repeats) sequences and alignment-free similarity measures developed previously [52]. The use of two identification methods ensured a more detailed search for hosts among prokaryotic organisms (VirHostMatcher-Net software) and identification of potential hosts among eukaryotic organisms (Virus–Host database).

For each of the performed identification methods, the count of the predicted viral scaffolds was transformed into tables representing DNA viruses that infect a certain host species.

### 2.11. Comparison of Viral Scaffold Taxonomic Composition, Predicted Hosts, and AMGs

Comparative statistical analysis of viral taxonomy, predicted host composition, and AMGs based on scaffold count tables was carried out by the NMDS ordination method with the algorithm described above.

## 3. Results

### 3.1. Identification and Taxonomic Annotation of Viral Reads

In this study, we obtained four virome datasets (V1–V4) from deep-water (pelagic zone) and shallow stations (area of the Maloye More Strait) of Lake Baikal (Table 1, Figure 1). After primary processing and filtering, our datasets contained from 2,218,572 to 3,573,602 reads, ranging from 100 to 301 bp (Table 3). The proportions of sequences identified as viral ones with significant matches, with the known sequences (e-value < 10^−5^, bit score ≥ 50) varied from 7.6 to 15.2% (168 557–423 054 of the reads); it was comparable to the number of viral reads in datasets from other freshwater ecosystems (Table 3).

Overall, 24 viral families of DNA viruses were revealed in the V1–V4 Baikal samples—19 families of dsDNA and 5 families of ssDNA viruses (Table 4; Appendix A). The families of bacteriophages, Siphoviridae (37.7–49.5% of reads), Podoviridae (26.9–52.0%), and Myoviridae (5.8–13.4%), were the most abundant. Other recently identified bacteriophage families of the order Caudovirales (Ackermannviridae and Herelleviridae) were minor (0.05–0.21% and 0.02–0.1%, respectively). The families Lavidaviridae, Phycodnaviridae, and Mimiviridae were also numerous (up to 2.4%). Totally, the listed dominant families accounted for more than 90% of the identified sequences. Some percentage of viral reads (2.89–7.67%) was similar to unclassified viruses (mainly to viruses of bacteria and archaea, Appendix A). The composition of the families in samples V1–V4 was the same, but the proportions of the families significantly differed. The water sample from the Maloye More Strait (V3) was the most distinctive: there was a much larger proportion of reads related to the Podoviridae and much fewer reads of Myoviridae compared to other samples. The smallest number of Phycodnaviridae and Mimiviridae was observed in sample V3, whereas the largest number was found in sample V1 (Table 4; Appendix A).

### 3.2. Composition of Virotypes in the Lake Baikal Datasets

Using selected options for taxonomic identification of viral reads (Section 2.5), we revealed 936, 937, 958, and 967 different virotypes in samples V1-V4, respectively (Table 3). The numbers of virotypes for the previously characterized Baikal samples, 6C, BVP1, and BVP2, were 958, 967, and 983, respectively. The Simpson diversity index for Baikal samples was more than 0.98, and the Shannon index ranged from 5.07 (for V3) to 5.6 (V4) (Table 3).

A large number and diversity of cyanophages that infected unicellular Cyanobacteria, *Synechococcus* sp., were among the most represented in all Baikal viromes (Figure 2; Appendix A). The virotypes of cyanophages, *Synechococcus phage S-CBS4*, *Synechococcus phage S-CBS3*, *Synechococcus phage S-CBS1*, and *Cyanophage KBS-S-2A* (representatives of the Siphoviridae family), dominated (>1% of virome reads). The myocyanophages were previously reported to be the most diverse and abundant in marine ecosystems [8,53]. In our study, cyanophages of the family Siphoviridae, including the recently isolated and characterized *Synechococcus* phages, *S-CBS1*, *S-CBS3*, and *S-CBS4* [54], were among the dominant ones in all viromes. The sequences similar to *Synechococcus phage S-SM2* of the family Myoviridae were abundant in sample V1 (1.03%) and the published 6C and BVP2 samples (6.96% and 1.4%, respectively) (Figure 2; Appendix A).

The viruses of the phylum Actinobacteria (*Arthrobacter phage Decurro*), Bacteroidetes of the class Flavobacteriia (for example, *Nonlabens phages P12024S* and *P1202L*, *Flavobacterium phage 11b*), Proteobacteria of the classes Alphaproteobacteria (*Puniceispirillum phage HMO-2011*, *Pelagibacter phage HTVC011P*, *Caulobacter phage Percy*, and *Aquamicrobium phage P14*), Betaproteobacteria (*Ralstonia phage RSK1* and *Rhodoferax phage P26218*), Gammaproteobacteria (*Idiomarinaceae phage 1N2-2* and *Xanthomonas phage Xp15*), and Verrucomicrobia (*Verrucomicrobia phage P8625*), as well as the archaeal virus, *Thermoproteus tenax virus 1*, were also in the list of dominant virotypes in the Baikal viromes (Figure 2; Appendix A). Despite almost the same composition, the distribution pattern of virotypes varied greatly in different Baikal samples (Figure 2). For example, in the composition of V3, we identified a significantly higher content of the virotypes infecting the genera *Caulobacter* (for example, *Caulobacter phage Percy*), *Aquamicrobium* (*Aquamicrobium phage P14*), *Pseudomonaceae* (*Pseudomonas phage PaMx42*), *Ralstonia* (*Ralstonia phage RSK1*, *Ralstonia phage RSB1*), and *Burkholderia* (*Burkholderia phage vB_BceS_KL1*) compared to other viromes. The sample BVP1 contained the highest number of *Thermoanaerobacterium phage THSA-485A* (Firmicutes, Clostridia) and virotypes infecting *Flavobacterium* sp. (Figure 2). Notably, many bacterial taxa that are potential hosts for identified virotypes (*Arthrobacter*, *Flavobacterium*, *Pseudomonas*, *Ralstonia*, *Caulobacter*, etc.) were previously determined in Baikal water using various approaches, including the culture-based method [55].

Eukaryotic viruses compared to bacteriophages were significantly less numerous in the studied datasets (<1%) but also diverse. Among the microalgae viruses of Phycodnaviridae (27 virotypes), representatives of the genera *Prasinovirus*, *Raphidovirus*, *Prymnesiovirus*, *Phaeovirus*, and *Chlorovirus*, as well as several other viruses unclassified at the rank of the genus, were present in samples V1-V4. *Chrysochromulina ericina virus* (0.1–0.56%), *Phaeocystis globosa virus* (0.05–0.29%), and *Bathycoccus sp. RCC1105 virus BpV1* (0.03–0.14%) were the most numerous virotypes of the family Phycodnaviridae (Appendix A).

The virotypes related to virophages *Yellowstone Lake virophages 5*, *6*, and *7*, which were obtained by de novo assembly of metagenomic shotgun sequencing databases from different locations of Yellowstone Lake [56] were among the most numerous in the Lake Baikal datasets, especially in the sample BVP1 (Figure 2; Appendix A). Previously, it was shown that virophages (Lavidaviridae family) are genetically diverse and widespread; they were found in various unique environments and geographical areas [56,57].

### 3.3. Analysis of Read Assemblies

As a result of metagenomic assembly using the SPAdes software, 4716 scaffolds were created with lengths from 5000 to 628,329 bp and coverage of more than 10× (Appendix A); of these, 1386 viral scaffolds with lengths from 5000 to 138,502 bp were identified with the VirSorter tool (Appendix A). Taxonomic affiliation (as virotype) was assigned for 1093 (78.9%) scaffolds (Appendix A). The resulting assembly, the viral scaffolds, and the predicted open reading frames (FASTA files) are available in the Figshare repository [58] (Appendix A).

Table 5 contains a set of viral scaffolds most represented in the Baikal viromes (the first five scaffolds arranged in descending order of the number of reads per scaffold in each sample were selected). The similarities of the predicted ORFs with those from the RefSeq proteome database averaged between 36.7 and 67.1% (Table 5). The dominant virotypes mainly included a large number of cyanophages of *Synechococcus* spp. and other bacteriophages, as well as the *Acanthocystis turfacea chlorella virus 1* and *Acanthamoeba polyphaga mimivirus*. As can be seen from Table 5, the composition of prevailing scaffolds varied in different samples. The longest scaffold, NODE_24_length_120194, that is closely related (in the number of similar proteins) to *Caulobacter phage Cr30* predominated in sample V1. The most abundant NODE_310_length_34107 (related to *Acinetobacter phage YMC11/11/R3177*), NODE_206_length_42308 (*Caulobacter phage Percy*), NODE_257_length_37045 (*Ralstonia phage RSK1*), and NODE_2600_length_10244 (not identified) prevailed to a large extent only in one or two samples. The NODE_424_length_29748 (attributed to *Pelagibacter phage HTVC008M*) were the most represented in all summer samples (V1-V4). It is noteworthy that the number of similar ORFs in this scaffold was extremely low (one of 39/40). The detected reading frames of NODE_2600_length_10244, NODE_4064_length_7673, and NODE_4202_length_7505, which mainly predominated in spring samples (BVP1 and BVP2, [28]), had no similarities with known viral proteins from the RefSeq database. In general, the number of identified proteins in the presented scaffolds was small, except for some scaffolds close to cyanophages (for example, NODE_696_length_22189 where the ratio of detected ORFs and predicted proteins was 25/21); the maximum identity (85.8 and 89.4%) of detected ORFs among dominant scaffolds was also observed with proteins of known cyanophage. This indicates a wide distribution of closely related viruses of picocyanobacteria and emphasizes the better study of the cyanophages in comparison with other bacteriophages inhabiting aquatic ecosystems.

### 3.4. Functional Analysis of the Baikal Viral Communities

In this study, we used an assembled dataset (Appendix A) for functional analysis of viral communities from different areas of Lake Baikal. The revealed groups of genes in the general set of assembled metavirome reads of Lake Baikal, according to the classification of COG and KEGG databases, are presented in Figure 3, Appendix A and Appendix A. The most representative functional categories, according to the COG database, included mobile genomic elements (prophages and transposons), the proteins and enzymes of cell wall/membrane/envelope biogenesis, as well as of replication, recombination, and repair, and unknown or only generally predicted functions. Structural viral proteins, such as portal, major capsid, tail tip/sheath, and other proteins, as well as enzymes, involved in the synthesis, modification, and packaging of DNA/RNA (polymerase, reductase, terminase, and helicase) were among the most numerous ones in the composition of the Baikal viromes (Appendix A). The KEGG pathway database revealed the genes of all five highest hierarchical categories, and the predominance of “metabolism” functions. The proteins of replication and repair (“genetic information processing” level), aging (”organismal systems”), signal transduction (“environmental information processing”), and cell growth and death (“cellular processing”) also prevailed in summary data on Lake Baikal. Listed categories predominated in all Baikal samples, but the ratio of other functional groups varied (Figure 3). Dendrograms based on the distribution of general functional categories demonstrated the similarity of the viral communities from deep-water columns in the pelagic zone of Southern and Central Baikal (V1 and V2), as well as of Listvennichny Bay (V4), and their difference from the shallow water samples from the Maloye More Strait area (V3) and the littoral zone of Southern Baikal (6C). Samples BVP1 and BVP2 (photic zone, early ice cover, and late spring sampling period, respectively) also differed from the other samples, similar to obtained taxonomic data (Figure 2).

The auxiliary metabolic genes (AMGs) from 11 secondary KO categories were defined by the VIBRANT program (Figure 4; Appendix A). Overall, 61 AMGs with different functions were identified in Baikal virome samples. Previously, [59] created the ”global ocean virome” dataset (of dsDNA viruses) and identified 243 viral-encoded auxiliary metabolic genes. The AMGs composition in both datasets partially overlapped; a total of 44 common genes were identified, but 21 genes of them were assigned to the category “others” (not to AMGs) in the study [59]. Furthermore, 17 genes were identified in the Baikal sets (such as *dcyD*, D-cysteine desulfhydrase; csn, chitosanase; *nadM*, nicotinamide-nucleotide adenylyltransferase, and others); however, it should be noted that for many of them no similarities were found with the known Pfam domains, according to which the identification of AMGs was carried out in [59]. Generally, most of the enzymes encoded by the identified AMGs are known to be present in the genomes of viruses.

The greatest number of functions, especially in samples V1–V4, was associated with the metabolism of carbohydrates (Figure 4; Appendix A). This was followed by the “amino acid metabolism” group, the most numerous in samples BVP1 and BVP2; and the “glycan biosynthesis and metabolism” category dominant in the sample V4. Samples 6C, BVP1 and BVP2 also showed a high content of AMGs of the “metabolism of cofactors and vitamins” group.

The percentage of individual AMGs also varied in different samples (Appendix A). The high content of gene *gltA* (citrate synthase) involved in the citrate cycle (“carbohydrate metabolism” category) was observed in samples V1–V4, especially in sample V3, where 25.4% of reads consisted of the scaffold containing this gene (NODE_424_length_29748, virotype *Pelagibacter phage HTVC008M*). This category also includes a high content of AMGs involved in the metabolism of propanoate (*mgsA*, methylglyoxal synthase; up to 8.8% of reads in sample V1, scaffold NODE_24_length_120194, virotype *Caulobacter phage Cr30*) (Table 5) as well as in pentose and glucuronate interconversions and pathway (*xylC*, xylonolactonase, and *pgl*, 6-phosphogluconolactonase; up to 6.7% in sample V4, NODE_225_length_40245, *Idiomarinaceae phage 1N2-2*). The genes of the fructose and mannose metabolism were the most abundant in sample 6C and identified in scaffolds assigned to cyanophages. For example, these are the genes *fcl* (GDP-L-fucose synthase, 4.7%, NODE_519_length_26092, *Synechococcus phage S-IOM18*), *gmd* (GDPmannose 4,6-dehydratase, 1.7%, NODE_481_length_27592, *Synechococcus phage S-SM2*), and *algA* (mannose-1-phosphate guanylyltransferase/ isomerase, 1.5%, NODE_266_length_36463, *Synechococcus phage S-SM2*) (Appendix A).

The identified enzymes from the category “glycan biosynthesis and metabolism”, are mainly involved in lipopolysaccharide biosynthesis (such as *waaC, waaF*, and *waaQ,* heptosyltransferases I-III, and others). The category “metabolism of cofactors and vitamins” is represented by the enzymes of the porphyrin and chlorophyll pathways (cobaltochelatases *cobS* and *cobT*), riboflavin (acid phosphatases *phoN* and *ACP5*), nicotinate, and nicotinamide metabolism (*nadM*; nicotinamide-nucleotide adenylyltransferase and others) as well as the folate (7-cyano-7-deazaguanine synthases *queC*, *queE*, and reductase *queF,* and others), ubiquinone and other terpenoid-quinone (*ubiG*; 2-polyprenyl-6-hydroxyphenyl methylase/3-demethylubiquinone-9 3-methyltransferase) biosynthesis. The enzymes classified as “energy metabolism” are associated with the sulfur pathway (*cysC*, adenylylsulfate kinase; *cysD*, sulfate adenylyltransferase subunit 2; and *cysH*, phosphoadenosine phosphosulfate reductase) and photosynthesis (*psbA*; photosystem II P680 reaction center D1 protein). The *psbA* gene in the genomes of cyanophages support the photosynthetic activity of infected cells, which in turn contributes to an increase in viral replication. The domains of phosphoadenosine phosphosulfate reductase (*cysH*) were regularly found in the genomes of viruses from pelagic freshwater habitats. This is one of the enzymes that are likely involved in the protection of their hosts from reactive oxygen species and in the oxidative burst in protist phagolysosomes [60].

### 3.5. Host Range Prediction

The potential host range (at the phylum level) for the revealed Baikal viruses (Figure 5A) was estimated, based on known hosts for the identified virotypes according to the Virus – Host database [50], and using the VirHostMatcher-Net software [51]. The former approach allowed us to estimate the putative hosts for a wide range of viruses (not only bacteriophages but also other viruses). Using this method, the putative hosts were determined for 1009 viral scaffolds (72.8%). In total, seven bacterial, one archaeal, and seven eukaryotic phyla, as well as one viral family (Mimiviridae affected by virophages during coinfection of protists), were identified (Figure 5B; Appendix A; Appendix A). Among the bacterial host taxa, the Cyanobacteria (15.3–51.2%), Proteobacteria (9.8–53.1%), Bacteroidetes (3.0–12.2%), Actinobacteria (5.0–10.8%), and Firmicutes (1.9–5.5%) prevailed; the Cyanobacteria dominated in samples V1, V2, V4, 6C, and BVP1, but the Proteobacteria—in V3 and BVP2. The percentage of Verrucomicrobia and Deinococcus_Thermus did not exceed 0.2% in different samples. The Euryarchaeota were from 0.1 to 1.1%. Among the eukaryotic hosts, the Chlorophyta were the most represented, accounting for 0.6 to 1.9%. The proportion of viral sequences with unidentified (unknown) hosts varied from 8.8% to 38.4% in different Baikal samples.

The host range predicted by VirHostMatcher-Net (intended for identification of prokaryotic hosts) included 20 bacterial and 3 archaeal phyla (Figure 5C; Appendix A; Appendix A); the putative hosts were defined for 1131 scaffolds (81.6%). The phyla Proteobacteria (20.5–57.9%), Firmicutes (7.5–38.3%), Bacteroidetes (10.5–41.4%), Actinobacteria (2.3–5.8%), and Cyanobacteria (1.6–4.4%) were the most numerous. Planctomycetes, Spirochaetes, and Acidobacteria were 0.4 to 2.9%, 0.3 to 3.0%, and 0.3 to 0.9%, respectively. The proportion of other phyla on average did not exceed 2.6%. Proteobacteria were the most numerous hosts in samples V1, V2, BVP1, and BVP2, whereas Firmicutes prevailed in samples V4 and 6C, and Bacteroidetes in sample V3. 

Although the number and ratio of bacterial host groups predicted by different methods varied, the same five microbial phyla dominated, which are Proteobacteria, Bacteroidetes, Cyanobacteria, Actinobacteria, and Firmicutes. Interestingly, the samples were clustered in the same way, according to the taxonomic affiliation of the viral scaffolds and AMGs profiles (Figure 5D), and based on their predicted hosts. The investigated Baikal viromes were divided into three group that included: (i) summer samples from this study (V1–V4), (ii) spring samples (BVP1-BVP2), and (iii) an autumn sample 6C from the littoral zone (Figure 5). Thus, a comprehensive analysis of the virome data clearly demonstrated the non-random nature of the distribution and formation of viral communities in the lake ecosystem.

### 3.6. Comparative Analysis of Viromes from Different Lake Ecosystems

The list of viral families identified in Lake Baikal (Table 4) and other lake ecosystems considered in this study (Table 2) were generally similar. In addition to the families listed in Table 4, we identified viruses of the families Parvoviridae (in all lakes, except for Baikal and Soyang) and Polydnaviridae (in small numbers in lakes Michigan and Ontario only) (Appendix A). Some families, especially those with a low number of reads, were absent in some lakes or samples. For example, reads close to Astroviridae viruses were revealed in all analyzed samples of Lake Michigan but only in one sample of Lake Soyang (Soyang_31) and most samples from Lake Baikal (except for samples V3 and 6C); Poxviridae were present in all viromes of Lake Baikal but only in some viromes of other lakes (Ontario_11, Erie_12, LoughN_17, Soyang_31, and Soyang_32). The virotypes of the Baculoviridae family were absent in Lake Biwa, in one Baikal sample (V3), and some samples from Lough Neagh (LoughN_16 and LoughN_21), etc. (Appendix A).

To compare the Baikal and other selected freshwater viromes in terms of the similarity of the taxonomic diversity (the composition and ratio of virotypes), we used hierarchical clustering and identified four clusters of the samples (Figure 6A). The first two clusters included the samples of Great Lakes: (1) Ontario_11; and (2) all samples of Lake Michigan. The third cluster (3) contained the samples Ontario_7, Ontario_9, Erie_12, Erie_14, Erie_15, and Baikal_V3. The fourth, most numerous, cluster (4) consisted of 21 samples from the lakes Baikal, Biwa, Soyang, and Lough Neagh. This cluster had several sub-clusters, one of those was formed by the samples from lakes Baikal, Biwa, and Soyang. Samples V1, V2, and V4 were the closest related among all Baikal samples; samples BVP1 and BVP2 were grouped separately. The separate sub-clusters composed all samples from Lough Neagh. The most distant samples in the fourth cluster were Biwa_40 and Soyang_29.

On the NMDS biplot (Figure 6B), the samples from cluster 4 formed a scattered group in the coordinate plane and partially intersected with cluster 3, specifically, samples Erie_12 and Baikal_V3 were closely located to cluster 4. Clusters 1 and 2 were the most distant from each other and clusters 3 and 4; accordingly, the samples of clusters 1 and 2 differed significantly in composition and virotype ratio from each other and the other clusters.

Figure 6 also shows the gradient vectors of ten dominant families in the samples, their species richness, and the Simpson and Shannon species diversity indices obtained in the NMDS analysis. Based on the direction of the gradient vectors, the samples of clusters 1 and 3 showed low values of species richness and the Simpson and Shannon indices, as well as high virotype counts of the families Phycodnaviridae, Myoviridae, Mimiviridae, and especially Parvoviridae and Bacilladnaviridae, and low virotype counts of the families Podoviridae and Siphoviridae. Cluster 2 had high virotype counts of the families Poxviridea and Inoviridae and low virotype counts of Lavidaviridae in contrast to the samples of cluster 4. The direction of the gradient vectors suggests that an increase in species richness and diversity is accompanied by an increase in the representation of viruses of the families Podoviridae and Siphoviridae and a decrease in viruses of the families Phycodnaviridae, Myoviridae, Mimiviridae, Parvoviridae, and Bacilladnaviridae in the samples.

On dendrogram and NMDS biplot, the Baikal samples, V1, V2, and V4, from the deep-water column taken during the same period were the most closely related, but sample V3 from the shallow area (Maloye More Strait) was distant. Among all available samples from Lake Baikal, viromes 6C and BVP1 were the most distant. The distinction of virome 6C from others is additionally associated with a specific sampling site (surface coastal waters) and season (November, biological autumn at Lake Baikal according to Kozhov [61]). Sample BVP1 from the photic layer of the pelagic zone (Listvyanka–Tankhoy section) was taken during early spring [61] in the under-ice period. Thus, cluster analysis, as well as data on taxonomic and functional analysis, likely represent both spatial and temporal differences of viral communities in Lake Baikal.

## 4. Discussion

In the present study, we investigated the composition and diversity of viral communities in Lake Baikal using a metagenomic approach. The new metavirome datasets provide an opportunity for a more complete assessment of the diversity and functional potential of viral communities of a large and ancient freshwater lake with unique characteristics and properties [19].

The bulk of viral sequences (99.3–99.9%) identified in our study were similar to dsDNA viruses. This predominance is associated with the library preparation technique for Illumina where, during linker ligation and amplification steps, dsDNA has an advantage [62]. Overall, only a small group of reads matched with single-stranded (ss) DNA viruses (0.10–0.62%). Recently, Roux et al. [63] have shown that ssDNA viruses are usually less abundant than dsDNA viruses in aquatic samples.

The three families of bacteriophages from the order Caudovirales—Siphoviridae, Podoviridae, and Myoviridae—were the most abundant in the Baikal samples (Table 4, Appendix A). Unexpectedly, the Siphoviridae family was dominant in samples V1, V2, and V4 (Table 4), as well as in previously published viromes from Lake Baikal (6C, BVP1, and BVP2), although the Myoviridae family dominated all samples of previous studies [27,28]. There are some causes of the change in the ratio of dominant families. Firstly, the RefSeq database is constantly enriching; in the present analysis, we used the updated version of early 2020. Secondly, as shown previously [27], the BLAST analysis parameters influence potential viral taxonomy; in this study, unlike previous studies [27,28], we used more strict conditions for the BLAST hit search. Thirdly, in our analysis, normalization to the length of the genome strictly shifted the read ratio towards the Siphoviridae family. For example, before normalization, the ratio of the Myoviridae and Siphoviridae families in samples V1 and 6C was 30.3% vs. 28.8% and 58.0% vs. 23.8%, respectively, but after normalization, the ratio became 13.7% vs. 40.2% and 29.6% vs. 43%, respectively (Appendix A).

The viruses closely related to known cyanophages of unicellular Cyanobacteria, mainly *Synechococcus* sp., predominated in all Baikal viromes (Figure 2; Appendix A). On the one hand, cyanophage dominance is a consequence of the high content of picocyanobacteria in Lake Baikal [64]; on the other hand, this is due to the great interest of researchers in cyanophages and the wide representation of their genomes in the NCBI RefSeq database. Hence, the proportion of cyanophages may be overestimated in our result. It should be noted that Proteobacteria and Actinobacteria are usually the most representative bacterial phyla in Lake Baikal as in other freshwater ecosystems [65,66], especially in deep-water samples; the contribution of the phylum Cyanobacteria to the analysis of the 16S rDNA genes is usually small (<10%) [27,67,68,69], except for the summer period (≥70%) [70]. In our previous study of the viral metagenome [27], cyanophages were also the most numerous in the sample; however, the contribution of Cyanobacteria to the bacterial community according to the analysis of the 16S rRNA gene was not great (only 8.7%). This indicates the bias of the resulting data towards well-studied viruses, as also mentioned previously [71].

It should be taken into account that most of the known cyanophages, as well as other viruses of aquatic origin, were isolated from marine ecosystems; therefore, the list of virotypes mainly includes the marine viruses (Appendix A). However, this situation is gradually changing, and the database of complete viral genomes is constantly enriched with freshwater viruses [32], including Cyanobacterial ones. Thus, in the future, bioinformatic analysis of available viral metagenomes from freshwater ecosystems will be more accurate.

In addition to a wide variety of bacteriophages, some hypothetical eukaryotic viruses were identified in our datasets. Among them, viruses of the families Phycodnaviridae (viruses of small algae), Mimiviridae (viruses of protists), and Poxviridae (viruses of invertebrates and vertebrates) were the most represented by the number of viral reads. The Phycodnaviridae viruses and the other giant nucleo-cytoplasmic large DNA viruses (NCLDVs) of the family Mimiviridae might be potentially affected by virophages [72] during co-infection of eukaryotic cells. The virophages (similar to *Yellowstone Lake virophages 5*, *6*, and *7*) also predominated in the Baikal viromes (Figure 2). *Yellowstone Lake phycodnaviruses 1*, *2*, and *3*, as well as *Yellowstone lake mimivirus* [73], were also the common virotypes in the Baikal and other freshwater viromes (Appendix A). We can assume the presence in the freshwater lakes of closely related eukaryotic hosts and their giant viruses that serve as the likely hosts for these virophages.

Based on the results of the taxonomic and functional analysis we revealed a high diversity of viral sequences, a greater similarity of viral communities of integral summer samples from deep-water columns (pelagic zone of southern and central basins and deep-water bay) of Lake Baikal and their difference from shallow areas (littoral zone and shallow strait), which is not surprising because of the significant difference in conditions and depth range of the stations. The littoral zone of the lake differs from the open pelagic area in more intensive biogeochemical processes, as high oxygen concentration, light penetration, and temperature conditions predetermine an increased content of organic matter and organism biomass [19,61]. The area of the Maloye More Strait, as mentioned above, has a special microclimate that is warmer than open Baikal. This is one of the most favorite and popular tourist places, leading to a high level of anthropogenic pollution, which most likely affects the composition of bacterioplankton and, accordingly, viral communities. Interestingly, the highest percentage of identified viral reads was observed for the samples taken from the area of the Maloye More Strait (maximum depth 25 m) and the coastal zone (sample 6C), i.e., in shallow areas: 15.2% and 19.5%, respectively (Table 3). We can assume that in the deep-water columns of Lake Baikal, in comparison with shallow ones, there are more unknown viruses that have no analogues in the NCBI RefSeq database.

In our study, we did not aim to trace seasonal variations or spatial distribution of viral communities (samples V1–V4 were taken at the same time and integral water samples collected from different depths were used); however, during the comparison of all known Baikal viromes, these trends are clearly observed. Most likely, in communities similar to bacterial [69,70,74], the greatest variety and the strongest seasonal changes in viral abundance and diversity occur in the upper layers of the lake where the temperature, light, and other factors play a significant role. However, the general processes that occur in the lake (deep convection, currents, seasonal developing, subsequent die-off, sedimentation of phyto- and bacterioplankton biomasses from the photic zone, and others) also make a certain contribution to the formation and changes of viral communities in the deep layers of the lake. The mixing processes in Lake Baikal determine the high diversity and some similarity of viral taxonomic composition throughout the lake; however, the local environmental conditions strongly affect the structure and diversity of plankton communities. Therefore, the viral communities in shallow and deep-water regions, as well as in the southern and central basins, also significantly differ (*p*-value < 0.05). A recent study [75] showed a vertical distribution and significant changes between epipelagic and bathypelagic Baikal viral community composition recovered from cellular metagenomes (fraction more than 0.22 μm). Accordingly, the difference between deep-water samples V1, V2, and V4 from other Baikal ones is partly explained by the presence of bathypelagic viral communities in them. However, based on the data on the greatest abundance of viruses at a depth of 0–25 m [76], we believe that in viromes from deep-water columns, the genomes of viruses from the photic surface layer of the lake water are the most represented and mainly determine the differences between samples.

Prediction of viral hosts based on metagenomic reads is a valuable tool for defining uncultured viruses, evaluation of virus–host interactions, the role of viruses in the regulation of various microbial populations, and the functioning of aquatic ecosystems in general. Using the chosen approaches, the different microbial taxa known as abundant in freshwater environment (Proteobacteria, Bacteroidetes, Cyanobacteria, Actinobacteria, Firmicutes, Verrucomicrobia, and others) were predicted as dominant viral hosts (Figure 5; Appendix A). These bacterial phyla were previously recorded based on the analysis of 16S ribosomal genes and shotgun sequencing [55,69,70,74,77]. Our results agree with the known data on the diversity of Baikal bacterial communities and demonstrate the presence of viruses infecting various taxonomic groups of microorganisms. Host prediction analysis, as well as taxonomic and functional (AMGs) analysis of viral scaffolds, revealed three groups of Baikal viromes, divided mainly by season (Figure 5). So far as the samples also differed in the depths and sites of sampling, it also most likely also influenced their clustering. Previously, the differences in the temporal (seasonal and interannual), vertical, and geographic distribution of microbial taxa in the pelagic and littoral zones of the lake were also shown [69,74,77]. Our results demonstrate a clear difference in composition of samples of both viral and microbial communities, and reflect the dynamics of these populations in Lake Baikal depending on habitat conditions. Unfortunately, it is not possible to identify the main factors of difference between the available viromes because the analyzed samples differ in many characteristics (sampling season, sampling site, depth range, etc.), and this issue requires more targeted studies. However, the results of taxonomic, functional and host prediction analyses are generally consistent and indicate that the distribution and diversity of Baikal viruses are not accidental and depend on various environmental factors.

Sampling for our study was carried out in early September (late summer, according to [61]) during the high abundance of phyto- (including blue-green algae) and bacterioplankton. Another peak of algal bloom and subsequent development of bacterioplankton occurs in March–April during the under-ice period. The different groups of diatoms and other algae dominate during these two periods [78]. Thus, we can assume that during these contrasting seasons, completely different groups of viruses of algae and metabolically related bacteria actively develop in the lake water, which is observed during a comparison of dsDNA viral communities (mainly bacteriophages) from spring (BVP1 and BVP2) and summer (V1–V4) Baikal samples. Unfortunately, viruses of eukaryotic phytoplankton (including diatoms) are less studied than prokaryotic ones. Furthermore, studies of RNA viruses, many of which affect phytoplankton, are necessary for a more complete comparative analysis of phytoplankton viruses. At present, the Refseq database contains 19 diatom viruses, all of which belong to RNA-containing viruses of the families Totiviridae, Flaviviridae, and Narnaviridae [32]. Among the DNA-containing viruses, the family Bacilladnaviridae also includes the viruses of diatoms. *Bacillariodnavirus LDMD-2013*, the genome of which is assembled from the marine virome [79], was revealed as the closest relative virotype of this family for all freshwater viromes.

The functional analysis of viral scaffolds revealed the different auxiliary metabolic genes involved in the cycles of carbon, nitrogen, sulfur, phosphorus, and their compounds (Figure 4; Appendix A); many of the identified AMGs have been previously found in viromes from aquatic environments (marine and freshwater) [59,60,80,81,82]. These metabolic genes stimulate the metabolic functions and defense mechanisms in host cells under various conditions, including rapidly changing and stressful ones, and, in general, contribute to the maintenance of the vital activity of the hosts and phage production (cyanophages, pelagiphages, and others) during infection. The sets of AMGs in the Baikal samples varied, as well as the diversity of viruses and their potential hosts. Thus, our results clearly demonstrate differences in the diversity and composition of viral communities, as well as in functional profiles, and rearrangements of their metabolic functions depending on environmental conditions.

Long-term studies of the ecosystem of Lake Baikal have shown the presence of stable seasonal dynamics of the phytoplankton community. As mentioned above, the bulk of the primary photosynthetic production of phytoplankton in Baikal is formed in the under-ice spring period (from March to May) due to the bloom of diatoms [19,83]. By the end of summer, the active phytoplankton growth ends; gradual cell death and active destruction of most of the accumulated organic matter by organotrophic bacteria occur [83]. In our study, the proportion of AMGs responsible for the metabolism of carbohydrates in the samples taken in the summer period (V1–V4) was significantly higher than in other samples (Figure 4). Thus, it is likely that during this period, dying phytoplankton provided organotrophic bacteria with a carbohydrate-rich substrate; and the proportion of viral species containing genes for carbohydrate metabolism increased. During infection, these phages stimulate the host metabolism and thereby accelerate the utilization of the carbohydrate substrate from the environment. In the samples from the ice, late spring and autumn periods (BVP1, BVP1, and 6C), the phages containing genes for the metabolism of complex substrates, amino acids, cofactors, vitamins, and nucleotides dominated in viral communities. In spring, phytoplankton is actively developing, and cell death has not yet occurred. In autumn the abundance of phytoplankton in the water is minimal; therefore, carbohydrates are not available in a large amount. Perhaps, in the spring and autumn seasons, the organotrophic bacterial community switches to the assimilation of amino acids, cofactors, vitamins, and nucleotides, and viral AMGs help them in these pathways.

Here, we also compared the Baikal virome data with those from other freshwater lakes. We used the same conditions of processing, identification, and description of virome reads for the correct comparison of different samples. It should be noted that we used a limited set of samples of freshwater lakes for our comparative analysis. In the future, we plan to expand the number of samples from various freshwater ecosystems and carry out a more complete and detailed comparison of freshwater viral communities. Based on our data, lakes Michigan, Ontario, and Erie were the most distinctive from Lake Baikal and other analyzed lacustrine ecosystems. Baikal samples (especially deep-water ones) tend to be closely related to lakes Soyang and Biwa, which is most likely due to the similarity of some hydrological parameters and a relatively close geographic location. Previously, we have indicated that the trophic status of lakes has a more significant effect on the structure of viral communities than their distance [24]. Therefore, we believe that the former factor is more important. Similar to the original studies [10,17] strong temporal dynamics were also observed in lakes Soyang (Korea) and Biwa (Japan) (Figure 6). In Lake Soyang, Soyang_29 (January) and Soyang_33 (September) were the most distinctive samples. Interestingly, the water quality in Lake Soyang strongly depends on monthly rainfall: during the monsoon period (July-August) the status of the water body changes from oligotrophic to high-mesotrophic and even low-eutrophic [84]. In ancient mesotrophic Lake Biwa, sample Biwa_40 (September, depth 65 m) was the most different. The wide NMDS biplot dispersion of samples from the same ecosystem indicated significant rearrangements in the structure of viral communities and their hosts in freshwater lakes, depending on environmental conditions. The smallest variations in viral communities were observed in Lough Neagh, all samples from which were closely related [16,18] (Figure 6). Lough Neagh is one of the well-known hypertrophic lakes and, unlike other lakes [66], it is characterized by low levels of Actinobacteria but the stable predominance of Cyanobacteria (in particular Planktothrix) [16,85].

Based on the environmental conditions, Lake Baikal is an extreme habitat; this explains the great variety of unique flora and fauna here. Comparative taxonomic analysis based on the identification of metagenomic reads using the NCBI RefSeq revealed a similar list of virotypes in Lake Baikal and other lakes (Appendix A). The question arises why the samples from Lake Baikal appeared to be close to other lakes (oligotrophic and mesotrophic). Organisms of Lake Baikal can be divided into two groups. The first group is typical Palaearctic flora and fauna, and the second group is endemics (endemic species, genera, and even families) [19]. Endemic species of Lake Baikal form the bulk of the biomass in the ecosystem [86]; endemic phytoplankton species provide a significant proportion of primary production [87,88]. The NCBI RefSeq database mainly contains viruses that infect widespread organisms of aquatic ecosystems (Palaearctic species). The viral communities of Lake Baikal (like other lakes) are in the initial phase of the study. The genomes of viruses that infect Baikal endemic species are not represented in the NCBI RefSeq database; thus, the DNA of unique Baikal viruses most likely has not yet been identified. In general, the variety and composition of viruses in Lake Baikal, especially in deep-water zones of the lake remains largely unexplored.

## 5. Conclusions

Here, we analyzed only several Baikal areas (the pelagic zone, the deep-water bay, and the shallow zone of the lake) and identified a high diversity and significant differences in the composition of their viral communities. Lake Baikal is characterized by various habitat conditions (different basins, bays and sors, a wide range of depths, strong seasonal changes, etc.). Therefore, future studies will require expansion of the range of stations, analysis of the vertical distribution of viral communities, identification of the relationship of viral and other planktonic communities, determination of the functional role of viruses in the ecosystem in more detail, etc.

Studies of freshwater viruses are relevant in light of global climate change, the accumulation of adverse anthropogenic factors for water bodies, and the deterioration of freshwater quality. Unfortunately, Lake Baikal is no exception. Currently, the lake is undergoing serious changes, which is associated with a climatic shift and an increase in anthropogenic pressure. The irreversible phenomena in Lake Baikal began after the construction of large industrial plants [89]; the signs of strong negative changes have been recording here since 2011 [90,91,92]. The highest eutrophication (changes in the structure and vertical zonation of benthic algae; the overgrowth of the lake bottom by filamentous green algae; mass development of species of the genus Spirogyra, which are normally not typical for the lake; the presence of the eutrophic diatom indicator *Fragilaria capucina* var. *vaucheriae*; disease and mass mortality of sponges; and other changes) is observed in the littoral zone [87,90,91,92,93]. The pelagic zone of the lake retains the characteristics of a highly oligotrophic reservoir; however, there is a tendency towards negative change [88]. Water samples for this study were taken in 2014, at the very beginning of observed changes in the ecosystem. Therefore, the viromes obtained in our study can be useful for future studies, especially for assessing the dynamics and long-term temporary changes in viral communities of the lake, and tracking the processes that occur in the coastal and pelagic areas of the lake.

## Figures and Tables

**Figure 1 microorganisms-09-00760-f001:**
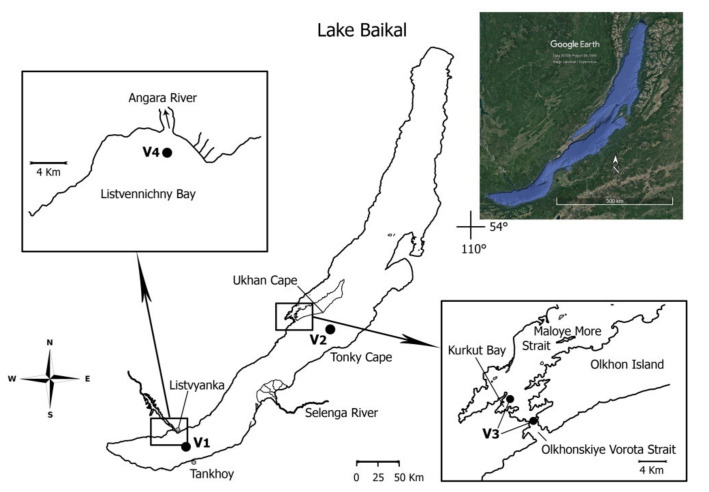
Lake Baikal and sampling sites (marked with a black circle). The lake shoreline contours and additional designations were reproduced using Google Earth Pro software (https://www.google.com/intl/ru/earth/versions/#earth-pro; accessed on 21 July 2020) with the free graphics editor “Inkscape” (https://inkscape.org/; accessed on 1 May 2020).

**Figure 2 microorganisms-09-00760-f002:**
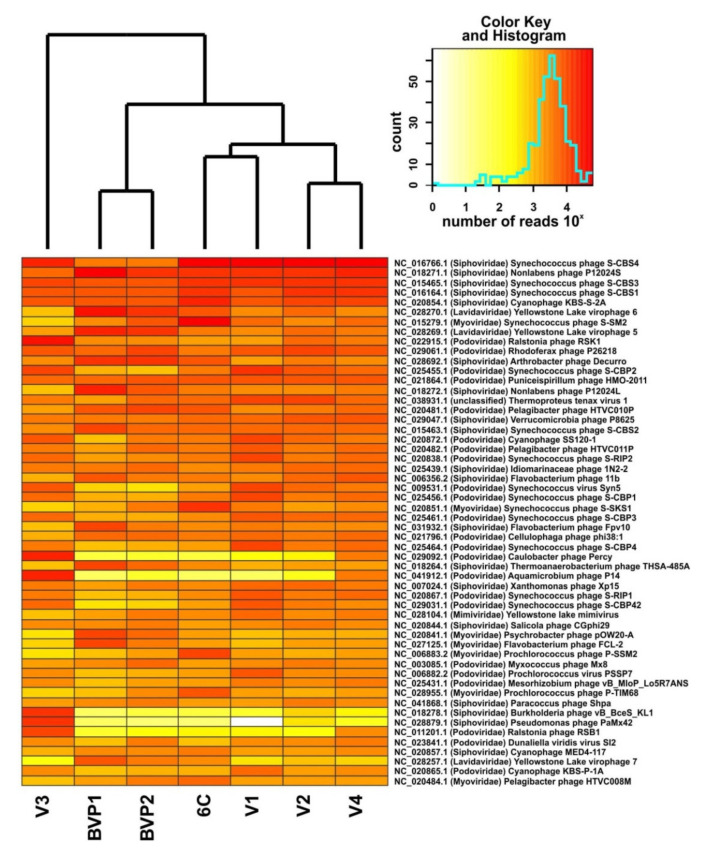
Heat map of the occurrence of the first 54 dominant virotypes comprising 50% of all sequences in the Baikal viromes. The list of published datasets is shown in Table 2. Read counts of virotypes are shown in log10 scale.

**Figure 3 microorganisms-09-00760-f003:**
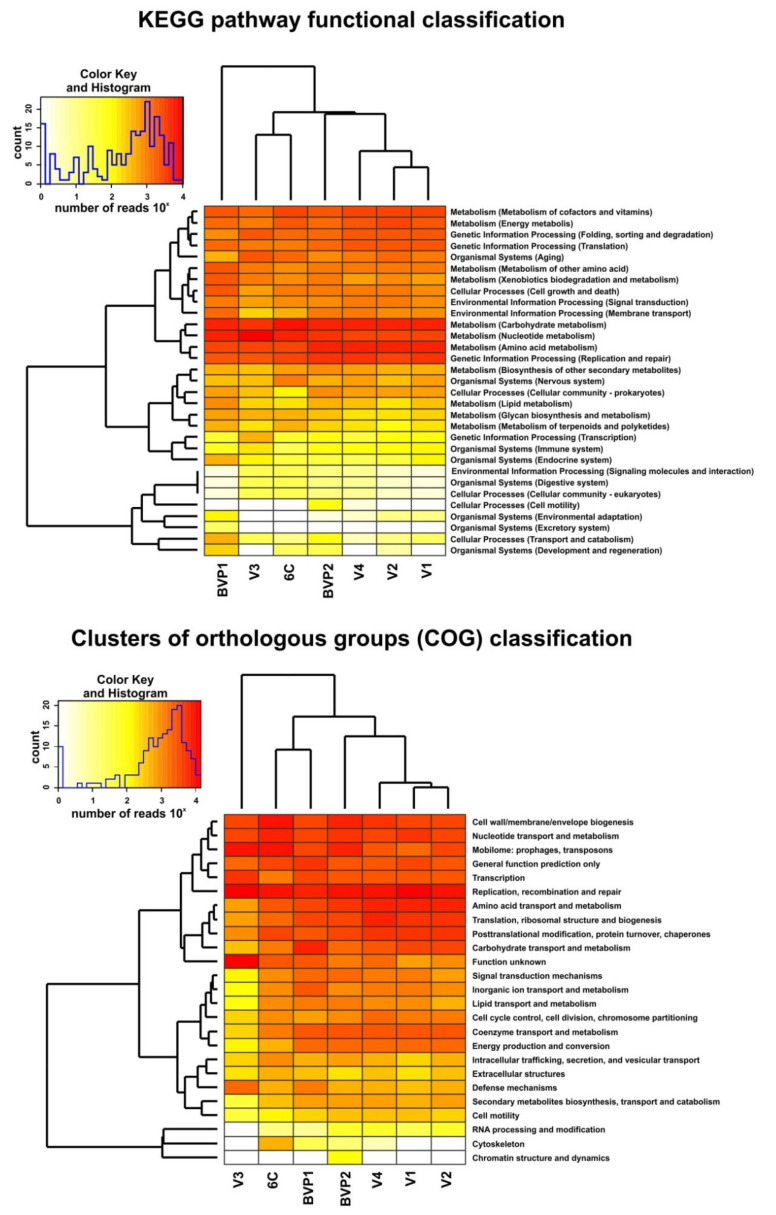
The main functional groups of the revealed genes according to the KEGG pathway and COG databases, and the clustering of the Baikal virome datasets based on functional analysis. Read counts of functional groups are shown in log10 scale.

**Figure 4 microorganisms-09-00760-f004:**
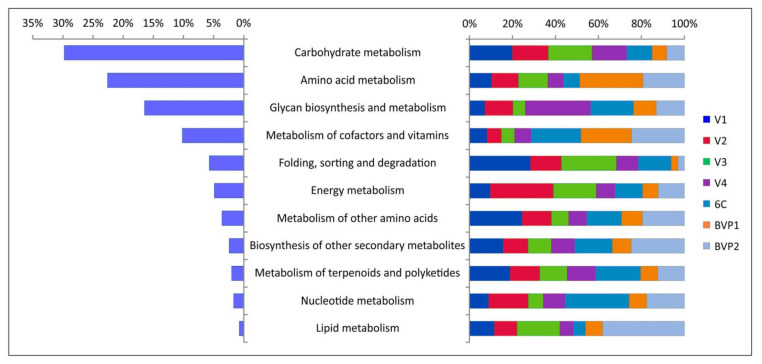
The main functional metabolic categories of auxiliary metabolic genes (AMGs) revealed in the virome datasets from Lake Baikal.

**Figure 5 microorganisms-09-00760-f005:**
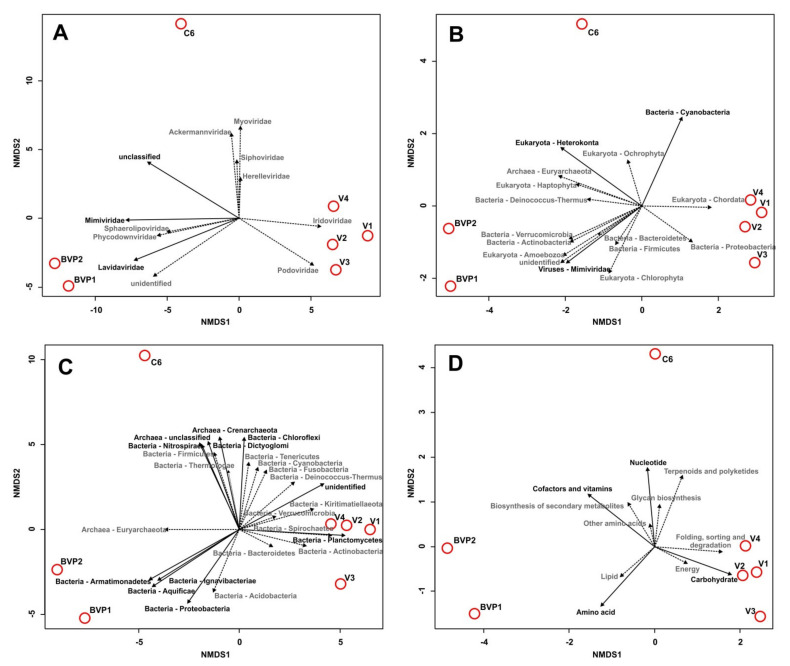
Non-metric multidimensional scaling (NMDS) biplots of the Baikal virome samples showing the following results: (**A**)—taxonomic analysis of viral scaffolds (vectors indicate the viral families); (**B**,**C**)—viral host prediction, carried out using the Virus–Host database and the VirHostMatcher-Net software, respectively (arrows show the phyla of putative host); and (**D**)—AMGs analysis (vectors are the metabolic categories). Unreliable vectors are marked with a dotted line and grey font.

**Figure 6 microorganisms-09-00760-f006:**
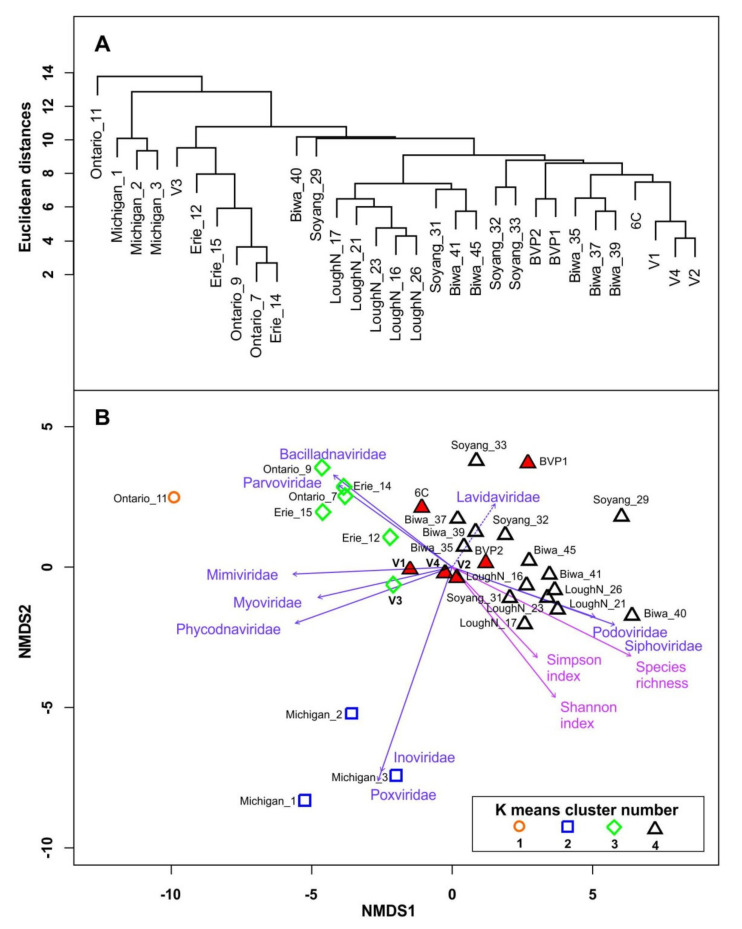
UPGMA (unweighted pair group method with arithmetic mean) cluster dendrogram (**A**) and NMDS biplot (**B**) showing the similarity of the samples (see Table 2) based on virotype counts (similarity of the taxonomic composition of viral communities). Blue arrows correspond to the vectors of gradient counts for ten dominant viral families in the samples (unreliable vector for the family Lavidaviridae is marked with a dotted line); pink arrows show biodiversity index gradients of viral communities; samples from Lake Baikal are marked in red.

**Table 1 microorganisms-09-00760-t001:** Sampling sites and their characteristics.

Sample	Station	Location	Date	Depth, m	Temperature, °C *
V1	Listvyanka–Tankhoy section, central site	51.704820 N, 105.010830 E	1 September 2014	0–500	12.60–3.41
V2	Uhan–Tonky section, central site	52.894300 N, 107.531940 E	3 September 2014	0–500	13.60–3.42
V3	Olkhonskiye Vorota Strait (a) and Kurkut Bay (b)	(a) 53.015175 N, 106.919555 E (b) 53.043735 N, 106.861745 E	3 September 2014	(a) 0–25 (b) 0	(a) 13.20–9.50 (b) N/A
V4	Listvennichny Bay	51.851853 N, 104.823476 E	5 September 2014	0–500	11.99–3.44

* Data of Laboratory of Hydrology and Hydrophysics, Limnological Institute, Siberian Branch of the Russian Academy of Sciences.

**Table 2 microorganisms-09-00760-t002:** Description of reference virome datasets used for analysis.

Dataset	Lake Name and Geographic Location	SRA Sample Name	Latitude and Longitude	Depth, m	Collection Date	Project	SRA Experiment	Illumina System	Layout	Reference
Baikal_6C	Lake Baikal, Russia	Lake Baikal_6C	51.899444 N 105.063889 E	0	8 November 2013	PRJNA398439	SRX3096544	MiSeq	PAIRED	[27]
Baikal_BVP1	Lake Baikal, Russia	Lake Baikal BVP1	51.798082 N 104.876782 E	0–50	22 March 2018	PRJNA547700	SRX5988169	MiSeq	PAIRED	[28]
Baikal_BVP2	Lake Baikal, Russia	Lake Baikal BVP2	51.82 N 104.9 E	0–50	8 June 2018	PRJNA547700	SRX5991721	MiSeq	PAIRED	[28]
Michigan_1	Lake Michigan, USA	Montrose 5-Jul-13	41.58 N 87.38 W	0	5 July 2013	PRJNA248239	SRX554904	MiSeq	PAIRED	[15]
Michigan_2	Lake Michigan, USA	Montrose 5-Jun-13	41.58 N 87.38 W	0	5 June 2013	PRJNA248239	SRX553227	MiSeq	PAIRED	[15]
Michigan_3	Lake Michigan, USA	Montrose 25-Jun-13	41.58 N 87.38 W	0	25 June 2013	PRJNA248239	SRX554914	MiSeq	PAIRED	[15]
Ontario_7	Lake Ontario, Canada	Lakeside-1_VD	43.20 N 79.27 W	1	9 August 2012	PRJNA288501	SRX1078016	HiSeq 2000	SINGLE	[14]
Ontario_9	Lake Ontario, Canada	Lakeside-3_VD	43.21 N 79.26 W	1	9 August 2012	PRJNA288501	SRX1077740	HiSeq 2000	SINGLE	[14]
Ontario_11	Lake Ontario, Canada	Lakeside-4_VD	43.20 N 79.27 W	1	4 July 2013	PRJNA288501	SRX1077738	HiSeq 2000	SINGLE	[14]
Erie_14	Lake Erie, Canada	Long Beach-1_VD	42.86 N 79.39 W	1	9 August 2012	PRJNA288501	SRX1077734	HiSeq 2000	SINGLE	[14]
Erie_12	Lake Erie, Canada	Long Beach-3_VD	42.87 N 79.42 W	1	9 August 2012	PRJNA288501	SRX1077730	HiSeq 2000	SINGLE	[14]
Erie_15	Lake Erie, Canada	Long Beach-4_VD	42.87 N 79.40 W	1	4 July 2013	PRJNA288501	SRX1077729	HiSeq 2000	SINGLE	[14]
LoughN_16	Lough Neagh, UK	4 pW	54.618333 N 6.395278 W	0–10	28 April 2014	PRJNA292054	SRX1134649	MiSeq	PAIRED	[16]
LoughN_17	Lough Neagh, UK	Lough Neagh, sample 1	54.618333 N 6.395278 W	0–10	4 March 2014	PRJNA350258	SRX2265652	MiSeq	PAIRED	[18]
LoughN_21	Lough Neagh, UK	Lough Neagh, sample 6	54.618333 N 6.395278 W	0–10	23 June 2014	PRJNA350258	SRX2267675	MiSeq	PAIRED	[18]
LoughN_23	Lough Neagh, UK	Lough Neagh, sample 8	54.618333 N 6.395278 W	0–10	2 September 2014	PRJNA350258	SRX2267700	MiSeq	PAIRED	[18]
LoughN_26	Lough Neagh, UK	Lough Neagh, sample 11	54.618333 N 6.395278 W	0–10	20 January 2015	PRJNA350258	SRX2267718	MiSeq	PAIRED	[18]
Soyang_29	Lake Soyang, South Korea	ERS2758845	37.947421 N 127.818872 E	0	January 2015	PRJEB15535	ERX2821557	MiSeq	PAIRED	[17]
Soyang_31	Lake Soyang, South Korea	ERS2759071	37.947421 N 127.818872 E	0	May 2016	PRJEB15535	ERX2821582	MiSeq	PAIRED	[17]
Soyang_32	Lake Soyang, South Korea	ERS2759121	37.947421 N 127.818872 E	0	November 2015	PRJEB15535	ERX2821583	MiSeq	PAIRED	[17]
Soyang_33	Lake Soyang, South Korea	ERS2759122	37.947421 N 127.818872 E	0	September 2015	PRJEB15535	ERX2821584	MiSeq	PAIRED	[17]
Biwa_35	Lake Biwa, Japan	LBV-julE	35.2193 N 135.9957 E	5	20 July 2016	PRJDB7309	DRX138423	MiSeq	PAIRED	[10]
Biwa_37	Lake Biwa, Japan	LBV-augE	35.2193 N 135.9957 E	5	18 August 2016	PRJDB7309	DRX138425	MiSeq	PAIRED	[10]
Biwa_39	Lake Biwa, Japan	LBV-sepE	35.2193 N 135.9957 E	5	27 September 2016	PRJDB7309	DRX138427	MiSeq	PAIRED	[10]
Biwa_40	Lake Biwa, Japan	LBV-sepH	35.2193 N 135.9957 E	65	13 September 2016	PRJDB7309	DRX138428	MiSeq	PAIRED	[10]
Biwa_41	Lake Biwa, Japan	LBV-octH	35.2193 N 135.9957 E	65	11 October 2016	PRJDB7309	DRX138429	MiSeq	PAIRED	[10]
Biwa_45	Lake Biwa, Japan	LBV-febH	35.2193 N 135.9957 E	65	13 February 2017	PRJDB7309	DRX138433	MiSeq	PAIRED	[10]

**Table 3 microorganisms-09-00760-t003:** General statistics and viral diversity indices for the datasets used in the study.

Sample	Data after Primary Processing			Data after Normalization and Filtration (95% Dominant Pool)					Reference
	Number of Reads	Number of Viral Reads (P) *	Percentage of Viral Reads	Number of Viral Reads (N + F) **	Species Richness	Shannon	Simpson	Chao1	
Baikal_V1	3,393,068	324,715	9.57	417,689	936	5.417	0.986	936	This study
Baikal_V2	2,218,572	168,557	7.60	250,497	937	5.583	0.986	937	This study
Baikal_V3	2,817,492	423,054	15.02	773,215	958	5.068	0.984	958	This study
Baikal_V4	3,573,602	296,176	8.29	411,218	967	5.573	0.987	967	This study
Baikal_6C	2,841,464	552,808	19.46	538,117	958	5.345	0.984	958	[27]
Baikal_BVP1	6,419,716	674,592	10.51	1,128,463	967	5.249	0.982	967	[28]
Baikal_BVP2	8,000,110	907,561	11.34	1,154,300	983	5.860	0.993	983	[28]
Michigan_1	3,519,620	158,971	4.52	175,007	877	5.948	0.995	877	[15]
Michigan_2	2,125,290	98,434	4.63	150,486	820	5.308	0.970	820	[15]
Michigan_3	2,966,796	122,085	4.12	147,116	899	5.962	0.995	899	[15]
Ontario_7	2,496,614	73,266	2.93	266,403	500	2.097	0.508	500	[14]
Ontario_9	2,822,912	99,697	3.53	200,576	509	3.529	0.844	509	[14]
Ontario_11	2,382,845	29,126	1.22	42,014	408	5.157	0.988	408	[14]
Erie_12	3,230,682	128,545	3.98	317,178	547	2.926	0.724	547	[14]
Erie_14	2,974,605	245,076	8.24	535,136	945	3.282	0.675	945	[14]
Erie_15	1,864,824	28,594	1.53	74,865	409	4.018	0.913	409	[14]
LoughN_16	4,559,044	567,073	12.44	950,193	966	5.482	0.986	966	[16]
LoughN_17	4,963,178	642,175	12.94	995,894	957	5.406	0.982	957	[18]
LoughN_21	3,808,264	522,507	13.72	903,063	955	5.470	0.986	955	[18]
LoughN_23	4,852,982	649,742	13.39	1,029,970	963	5.657	0.989	963	[18]
LoughN_26	7,016,230	851,187	12.13	1,387,862	971	5.536	0.984	971	[18]
Soyang_29	7,999,738	806,148	10.08	1,316,771	967	5.594	0.987	967	[17]
Soyang_31	8,000,708	946,185	11.83	1,579,639	981	5.383	0.982	981	[17]
Soyang_32	8,002,442	1,051,361	13.14	1,381,170	981	5.579	0.985	981	[17]
Soyang_33	8,002,316	1,230,079	15.37	1,423,692	967	5.587	0.988	967	[17]
Biwa_35	6,743,582	697,395	10.34	1,214,025	951	4.910	0.960	951	[10]
Biwa_37	6,686,984	719,098	10.75	1,148,642	947	5.049	0.977	947	[10]
Biwa_39	7,246,780	674,943	9.31	1,037,938	949	5.356	0.986	949	[10]
Biwa_40	2,540,588	197,093	7.76	323,135	923	5.707	0.992	923	[10]
Biwa_41	8,000,012	650,161	8.13	1,078,176	959	5.468	0.984	959	[10]
Biwa_45	2,770,592	203,250	7.34	350,545	919	5.441	0.986	919	[10]

* P—Number of viral reads after primary processing; ** N + F—Number of viral reads after normalization to the length of genome and filtration (95% dominant pool).

**Table 4 microorganisms-09-00760-t004:** The viral families identified in the studied and published viromes [27,28] from Lake Baikal.

Family	Type	Known Hosts	V1	V2	V3	V4	6C	BVP1	BVP2	Total
Siphoviridae	dsDNA	bacteria	40.14	49.54	37.70	45.99	42.53	56.57	47.61	320.08
Podoviridae	dsDNA	bacteria	36.49	26.90	52.04	28.42	16.80	14.54	22.06	197.25
Myoviridae	dsDNA	bacteria	13.44	13.37	5.79	12.90	30.33	11.45	15.62	102.91
Lavidaviridae	dsDNA	protists infected by mimivirus	1.11	2.39	0.57	1.04	3.57	12.73	7.48	28.88
unclassified	mainly dsDNA	-	3.93	4.12	2.89	7.67	3.45	2.32	2.42	26.79
Phycodnaviridae	dsDNA	algae	2.26	1.65	0.49	1.75	1.65	0.78	1.88	10.47
Mimiviridae	dsDNA	protists	0.96	0.57	0.15	0.75	0.50	0.33	0.78	4.04
Bacilladnaviridae	ssDNA	diatoms	0.11	0.29	0.08	0.29	0.26	0.22	0.56	1.81
Poxviridae	dsDNA	vertebrate, arthropoda	0.38	0.23	0.02	0.26	0.04	0.05	0.28	1.26
Inoviridae	ssDNA	bacteria	0.10	0.17	0.00	0.08	0.12	0.14	0.37	0.97
Ackermannviridae	dsDNA	bacteria	0.21	0.15	0.06	0.12	0.18	0.07	0.12	0.91
Herelleviridae	dsDNA	bacteria	0.10	0.09	0.03	0.11	0.11	0.08	0.12	0.64
Sphaerolipoviridae	dsDNA	bacteria, archaea	0.04	0.07	0.04	0.03	0.09	0.11	0.15	0.53
Totiviridae	dsRNA	fungi, protists	0.00	0.00	0.00	0.00	0.00	0.33	0.05	0.39
Astroviridae	ssRNA	vertebrates	0.25	0.07	0.00	0.03	0.00	0.01	0.02	0.37
Iridoviridae	dsDNA	amphibia, insects, fish	0.06	0.06	0.01	0.05	0.08	0.03	0.07	0.36
Bicaudaviridae	dsDNA	archaea	0.07	0.05	0.01	0.08	0.03	0.03	0.05	0.32
Microviridae	ssDNA	bacteria	0.00	0.00	0.00	0.18	0.02	0.03	0.03	0.25
Tobaniviridae	ssRNA	vertebrate	0.05	0.04	0.02	0.04	0.02	0.02	0.04	0.24
Baculoviridae	dsDNA	arthropods	0.04	0.05	0.00	0.04	0.02	0.03	0.04	0.23
Potyviridae	ssRNA	plants	0.04	0.03	0.02	0.01	0.05	0.03	0.03	0.21
Papillomaviridae	ssDNA	vertebrate	0.07	0.05	0.01	0.04	0.01	0.00	0.03	0.21
Nudiviridae	dsDNA	insects, crustacea	0.06	0.04	0.01	0.04	0.01	0.01	0.03	0.19
Spiraviridae	ssDNA	archaea	0.03	0.01	0.01	0.02	0.03	0.03	0.04	0.18
Tectiviridae	dsDNA	bacteria	0.02	0.02	0.03	0.01	0.01	0.01	0.04	0.16
Caulimoviridae	dsDNA	plants, insects	0.00	0.00	0.00	0.01	0.06	0.02	0.03	0.13
Ascoviridae	dsDNA	insects	0.02	0.02	0.00	0.01	0.00	0.01	0.02	0.10
Adenoviridae	dsDNA	vertebrate	0.01	0.01	0.00	0.01	0.01	0.01	0.01	0.06
Herpesviridae	dsDNA	vertebrate	0.01	0.01	0.00	0.01	0.00	0.02	0.01	0.05

**Table 5 microorganisms-09-00760-t005:** The viral scaffolds mostly represented in the Baikal samples (including the published ones [27,28]); the maximum and average similarity (in %) of predicted viral proteins with the NCBI RefSeq viral proteome database and related virotypes (the five largest sets of reads corresponding to specific virotype in each sample are marked in bold).

Scaffolds	Detected/Predicted ORFs	Max Similarity	Average Similarity	Virotype	RefSeq_ID	V1	V2	V3	V4	6C	BVP1	BVP2
NODE_24_length_120194	107/18	60.3	40.7	*Caulobacter phage Cr30*	NC_025422	**5257**	2958	2668	3382	3497	3082	6846
NODE_49_length_86592	47/4	55.2	45.9	*Acanthocystis turfacea chlorella virus 1*	NC_008724	483	2557	394	**30,145**	1394	5429	5848
NODE_50_length_85337	80/3	42.3	38.5	*Klebsiella phage Sugarland*	NC_042093	1397	**5436**	2179	**29,768**	**7350**	958	1127
NODE_86_length_73027	123/10	49.4	41.8	*Arthrobacter phage Laroye*	NC_041947	698	**6710**	7623	2578	236	12	16
NODE_124_length_59027	64/6	56.8	46.6	*Azospirillum phage Cd*	NC_010355	2	39	22	4	**6958**	15	18
NODE_206_length_42308	50/21	60.3	45.9	*Caulobacter phage Percy*	NC_029092	6	151	**226,165**	**20,294**	6	1	0
NODE_252_length_37409	32/28	59.1	67.1	*Synechococcus phage S-SKS1*	NC_020851	657	118	91	381	**7587**	687	1819
NODE_257_length_37045	44/31	82.5	56.1	*Ralstonia phage RSK1*	NC_022915	0	1300	**129,053**	517	1	0	3
NODE_310_length_34107	52/10	78.4	47.0	*Acinetobacter phage YMC11/11/R3177*	NC_041866	10	0	5	10	8	17	**125,574**
NODE_312_length_34016	54/16	62.0	50.2	*Bacillus phage TP21-L*	NC_011645	8	13	6	5	7	8	**40,371**
NODE_393_length_30742	40/1	77.3	36.7	*Mycobacterium phage 40AC*	NC_023607	5458	**9345**	17,380	11,664	746	550	425
NODE_417_length_29904	43/10	36.7	45.7	*Acinetobacter phage phiAC-1*	NC_028995	6839	**11,253**	**19,135**	10,041	2803	7288	7206
NODE_424_length_29748	39/1	78.3	57.0	*Pelagibacter phage HTVC008M*	NC_020484	**13,391**	**8793**	**24,103**	**13,626**	2036	1153	1194
NODE_461_length_28257	39/16	57.1	39.9	*Nonlabens phage P12024L*	NC_018272	**13,305**	171	349	319	295	324	597
NODE_630_length_23488	36/4	67.3	50.9	*Cellulophaga phage phi46:3*	NC_021792	273	808	516	1129	1361	**54,245**	**13,176**
NODE_670_length_22640	20/16	39.6	51.2	*Synechococcus phage S-CBS4*	NC_016766	**9496**	4122	13,239	4213	3854	207	233
NODE_696_length_22189	25/21	65.4	60.1	*Synechococcus phage S-CBS4*	NC_016766	3703	2206	4370	3133	**8250**	443	495
NODE_841_length_19701	23/16	73.9	54.2	*Synechococcus phage S-CBP2*	NC_025455	**17,988**	5282	**24,670**	4705	1701	553	332
NODE_857_length_19451	36/21	85.8	52.7	*Synechococcus phage S-CBS4*	NC_016766	1081	480	1114	743	**6190**	352	505
NODE_862_length_19396	23/8	63.4	47.8	*Acanthamoeba polyphaga mimivirus*	NC_014649	223	552	537	**13,041**	266	16	17
NODE_2600_length_10244	12/0	53.1	NA	NA	NA	28	26	25	15	14	**219,170**	**60,924**
NODE_2821_length_9717	10/1	81.6	39.9	*Thermoanaerobacterium phage THSA-485A*	NC_018264	419	656	381	410	1772	**32,617**	8577
NODE_4064_length_7673	6/0	NA	NA	NA	NA	349	457	495	144	588	**48,596**	**14,868**
NODE_4094_length_7637	7/3	89.4	39.5	*Cyanophage NATL1A-7*	NC_016658	**9762**	425	3540	1096	167	35	25
NODE_4202_length_7505	7/0	39.9	NA	NA	NA	167	227	227	62	305	**31,325**	12,258

## Data Availability

Unprocessed virome reads for samples V1–V4 from Lake Baikal were submitted to the NCBI SRA database (BioProject PRJNA398439, BioSamples SAMN15770410-SAMN15770413). The data presented in this study (as Appendix A), including the count tables on the taxonomic, functional and hosts prediction analysis (Appendix A), resulting assembly, viral scaffolds and proteins in fasta format (Appendix A), and additional figures (Appendix A) are openly available in the FigShare repository at https://dx.doi.org/10.6084/m9.figshare.12814637; accessed on 3 April 2021, reference number [58].

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
