# Peer review of "Extended Evaluation of Viral Diversity in Lake Baikal through Metagenomics"

_microorganisms, 2021, doi:10.3390/microorganisms9040760_

Round 1

Reviewer 1 Report

This study investigates viral diversity in Lake Baikal, by sequencing viromes from four different sites (3 deeper sites and 1 shallow site). Samples from different depths at each site were combined, and the integrated samples were sequenced. Viromes in Lake Baikal have been studied previously and published by the same research group. They intended to include more viromes from broader sites in this study. The authors also compared their samples with viromes from other freshwater systems, and this certainly makes the paper more interesting. I believe that additional comparison with estuarine and marine viromes would be helpful to the paper, especially the marine system was mentioned in the introduction. The structure of manuscript needs to be improved. In many places, discussions were carried out in the results section. Some specific comments are as follows.

L61: Winter condition should be described briefly, i.e. ice, water temperature, etc.

L69: Any major difference between the lake and marine systems based on the g20 and g23 gene markers?

L75-77: and the outcome of this study? Briefly

L79: I expect that the authors would compare viromes between the lake and marine water (just a few samples), to broaden the impact. The comparison between the freshwater and marine system was mentioned in the early section of introduction.

L80-94: Move this portion to Materials and Methods.

For Materials and Methods, more details should be provided for water sample collection and combination. How much water was collected at each depth? How were they combined and what is the final volume? 20L was mentioned earlier, but it is not clear this is the final volume used for concentrating viral particles. Integration of samples from different depths need some justification because viral communities vary along the depth profile. Pros and cons of combining samples from different depths at each site should be discussed.

L277-282: This part belongs to discussion.

L299-310: Reads like discussion.

L316: define virotypes, i.e. length of viral contigs, and how was each virotype determined?

L334-345: This part reads like discussion

L346-352: Discussion

L383-394: There seems to be a mixture of results and discussion here.  At the beginning of paragraph, the authors should report what your data tell us.

L640-655: This section is more like a background, and is not tied to your results.

Author Response

Dear Reviewers,

We are sincerely grateful to you for careful attention and critical comments to our study, which greatly improved the quality of our manuscript. We conducted an additional analysis of the data, carefully edited the text of the manuscript and checked the English again. The sentences, paragraphs and sections that we have edited and supplemented are marked with a yellow background in the text of manuscript. Our comments are listed below.

Review 1

This study investigates viral diversity in Lake Baikal, by sequencing viromes from four different sites (3 deeper sites and 1 shallow site). Samples from different depths at each site were combined, and the integrated samples were sequenced. Viromes in Lake Baikal have been studied previously and published by the same research group. They intended to include more viromes from broader sites in this study. The authors also compared their samples with viromes from other freshwater systems, and this certainly makes the paper more interesting. I believe that additional comparison with estuarine and marine viromes would be helpful to the paper, especially the marine system was mentioned in the introduction. The structure of manuscript needs to be improved. In many places, discussions were carried out in the results section. Some specific comments are as follows.

L61: Winter condition should be described briefly, i.e. ice, water temperature, etc.

  • It was added: “Lake Baikal freezes in the first half of January (ice thickness up to 130 cm); the ice breaks in early May. During the warm period (July to September) the temperature of the surface water layer in open Baikal is 5 to 19°С, and the temperature of the deep-water layer is ca. 4°С. [20]”.

L69: Any major difference between the lake and marine systems based on the g20 and g23 gene markers?

  • It was added: “Previous studies [25, 26] demonstrated through UniFrac method that g23 and g20 assemblages from freshwater lakes, including Baikal ones, were more closely related to those from terrestrial aquatic environments (wetlands, paddy fields, and upland soils) than to those from marine ones”.

L75-77: and the outcome of this study? Briefly

  • The main findings of this study were added: “Recently, Potapov et al. [28] examined two more viromes from the photic layer of the pelagic zone of the lake during the under-ice and late-spring periods and revealed some differences in the taxonomic and functional composition of viruses in these datasets. A comparative analysis of viral communities from different environments indicated a distribution pattern by soil, marine and freshwaters. Baikal viromes formed a separate clade with those from the Great Lakes of North America (Michigan, Erie, and Ontario)”.

L79: I expect that the authors would compare viromes between the lake and marine water (just a few samples), to broaden the impact. The comparison between the freshwater and marine system was mentioned in the early section of introduction.

  • In this study, we did not aim to compare the viral communities of Lake Baikal with marine ones because the difference between freshwater viruses, including the Baikal ones, and those from other environments, including marine ones, was shown in previous studies based on marker genes and metagenomic analysis. This information and references were added (Lines 64-67, 74-77).

Moreover, according to our approach, all datasets from lake ecosystems were processed together at the first stage of analysis. The addition of marine viromes will require a complete rework of the taxonomic and subsequent analysis, which will take a lot of time and more resources; but the time provided by Editors to revise the manuscript was extremely limited.

L80-94: Move this portion to Materials and Methods.

  • The text about sampling was moved to Materials and Methods and combined with the section 2.1. “Description of study sites and sample processing”. The hypotheses tested and main conclusions were formulated here according to the Instructions for Authors of the journal: “It (Introduction) should define the purpose of the work and its significance, including specific hypotheses being tested… Finally, briefly mention the main aim of the work and highlight the main conclusions” (Lines 80-87).

For Materials and Methods, more details should be provided for water sample collection and combination. How much water was collected at each depth? How were they combined and what is the final volume? 20L was mentioned earlier, but it is not clear this is the final volume used for concentrating viral particles. Integration of samples from different depths need some justification because viral communities vary along the depth profile. Pros and cons of combining samples from different depths at each site should be discussed.

  • We revised this paragraph and tried to describe the process of collecting and processing samples in more detail. The pros and cons of combining samples from different depths at each site were also briefly discussed (Lines 110-133).

L277-282: This part belongs to discussion.

  • This part was moved to Discussion (Lines 645-650).

L299-310: Reads like discussion.

  • This part was moved to Discussion (Lines 653-665).

L316: define virotypes, i.e. length of viral contigs, and how was each virotype determined?

  • The virotypes presented in the manuscript had been identified based on the taxonomic identification of reads. We clarified this in the text (Line 358). The identification of virotypes was described in more detail in Materials and Methods, in section 2.5., the title of which we have also edited (from "Taxonomic analysis" to "Taxonomic analysis of viral reads").

L334-345: This part reads like discussion

  • This part was moved to Discussion (Lines 666-680).

L346-352: Discussion

  • This part was moved to Discussion (Lines 681-687).

L383-394: There seems to be a mixture of results and discussion here.  At the beginning of paragraph, the authors should report what your data tell us.

  • This paragraph was revised and partly moved to Discussion (Lines 688-699).

L640-655: This section is more like a background, and is not tied to your results.

  • We agree that this section is not tied to our results. Here, we raise the topical issue about recent serious changes in the ecosystem of Lake Baikal that indicates the prospects for further research and emphasizes the importance of our data. Now, we have moved this paragraph with some additions to the section “Conclusions” (Lines 866-883).

Best regards,

Tatyana V. Butina and Yurij S. Bukin

Reviewer 2 Report

Paper consistent, thorough and well prepared. Below are some comments.

To the whole paper: I do not support writing in the form "we"; I believe that impersonal forms should be used in scientific paper.

L78-86: too much info, especially on the methodology; this excerpt is meant to give only the aim and main objectives of the research.

L86-94: this passage should not be in the introduction; it can be in the abstract, but the summary of results cannot be in the introduction. Please reword.

Table 1: I suggest moving to methodology, to the study site description.

L98: I would break it into 2 subsections: e.g. (1) study site and sample processing; (2) DNA extraction. And then Table 1 should be next to the description of the site/field, then a description of how samples were collected, when, in what quantity, repetitions, etc. And in the next subsection a description of DNA isolation and Table 2.

L115: explain the abbreviation VLPs

L245: Which R is it? R studio? Provide references.

L299-307: this is more of a discussion; not a result

L337-345; 385-393: this is more of a discussion; not a result

L656-663: this should probably already be signed as "conclusions".

Figure S2: legend should be captioned (what taxonomic level it presents); no unit for graphs.

Table S: The supplement tables would look better in excel. At the moment some have very unreadable (overlapping) columns.

Author Response

Review 2

Paper consistent, thorough and well prepared. Below are some comments.

To the whole paper: I do not support writing in the form "we"; I believe that impersonal forms should be used in scientific paper.

  • Previously, we also held the same opinion. However, in recent years in scientific literature, the use of the personal form is very common, and some journals even recommend it in the Instructions for Authors. This form is very convenient for presenting and interpreting our results as opposed to published ones. We consulted with a professional translator and he confirmed our opinion and noticed that the use of the personal form is quite appropriate and not abused in our manuscript. Therefore, if you and the Editors will allow, we will leave the use the form with "we" in this manuscript.

L78-86: too much info, especially on the methodology; this excerpt is meant to give only the aim and main objectives of the research.

  • We shortened this paragraph. The text about sampling was moved to Materials and Methods and combined with the section 2.1. “Description of study sites and sample processing”.

L86-94: this passage should not be in the introduction; it can be in the abstract, but the summary of results cannot be in the introduction. Please reword.

  • We slightly edited this text, leaving only our working hypotheses and the main conclusions, according to the requirements of the journal: “It (Introduction) should define the purpose of the work and its significance, including specific hypotheses being tested… Finally, briefly mention the main aim of the work and highlight the main conclusions” (Lines 80-87).

Table 1: I suggest moving to methodology, to the study site description.

  • It was moved to Materials and Methods

L98: I would break it into 2 subsections: e.g. (1) study site and sample processing; (2) DNA extraction. And then Table 1 should be next to the description of the site/field, then a description of how samples were collected, when, in what quantity, repetitions, etc. And in the next subsection a description of DNA isolation and Table 2.

  • Two recommended subsection were done. We also moved part of the text (about sampling sites, Lines 93-97) and the Figure 1 from the Introduction.

L115: explain the abbreviation VLPs

  • It was done: “VLPs (virus-like particles)” (Line 117).

L245: Which R is it? R studio? Provide references.

  • It was specified. The full name of the R (“the R Project for Statistical Computing”) and the reference were added.

L299-307: this is more of a discussion; not a result

  • This part was moved to Discussion (Lines 653-665).

L337-345; 385-393: this is more of a discussion; not a result

  • These parts were also included to Discussion (Lines 666-680, 688-699).

L656-663: this should probably already be signed as "conclusions".

  • This was included in the Conclusions section.

Figure S2: legend should be captioned (what taxonomic level it presents); no unit for graphs.

  • The legends were added to all Supplementary figures.

Table S: The supplement tables would look better in excel. At the moment some have very unreadable (overlapping) columns.

  • The tab-delimited format used in our manuscript to save the analysis data is generic and appears as a spreadsheet when opened in Microsoft Excel and other spreadsheet programs (Windows, Linux, and macOS). Moreover, the tab-delimited format will be convenient for those who use the pipelines for data processing written in programming languages (R, Phyton, and alike ones).

Reviewer 3 Report

In this manuscript the authors evaluated the diversity of viruses at four sites from lake Baikal through metaviromics. This is a relevant topic with major knowledge gaps. Therefore, this work contributes to ongoing efforts to described the viral diversity of this ecosystem and the overall microbial diversity of freshwater habitats. The experiments are well designed and the results support the majority of conclusions. Nevertheless some issues must be addressed to make this manuscript suitable for publication

Major issues:

The viromes are derived from integrated samples from multiple depths. This is expected to significantly impact the results because the viral diversity is know to change with depth. Although this does not invalidate the results this limitation needs to be considered when interpreting the results and highlighted in the discussion.

There is a lack of clarity in the methodology (detailed below) which make it particularly difficult to interpret the results.

Lack of relevant host information. No effort has been made to assign hosts to the assembled viral sequences. This is the most relevant question about the biology of these viruses and there are several tools available approaches achieve this in silico [1–5]⁠.

Lack of relevant information about viral metabolic potential. Authors analysed the functional roles of viral proteins only superficially. Auxiliary metabolic genes, an important aspect of viral communities with some novel examples from lake Baikal [6]⁠, have not been analysed properly. This data is also fundamental for a proper understanding of viral contribution to ecosystem functioning and there are also tools designed to identify such genes in silico [7,8]⁠. Coupled with host prediction data, exploring the diversity of auxiliary metabolic genes would significantly enhance the relevance of this manuscript.

Minor issues:

Ln 39-40: Not clear what is meant by wide and deep geography

Ln 43-45: Sentence too long, verbs and commas missing.

Ln 69-70: Please elaborate: How exactly the environmental factors influence these viral communities?

Ln 80 “viral shotgun metaviromes” is a redundancy, use “viral metagenomes” or “metaviromes”

Ln 161-164: Confusing sentence, plus the concept of virotype is not properly introduced.

Ln 152-174: This section is confusing and difficult to follow. It is unclear to me how the abundances based on genomic and proteomic data were combined. Also, there is no reference to this particular methodology being used before. It the authors want to introduce a new method it is important to describe it more clearly, justify its choice and demonstrate its superiority over more classical approaches of calculating viral abundances based on metagenomic data (such as percentage of mapped reads or RPKM)

Ln 192: Unclear what is meant by hidden diversity

Ln 199-201: This type of unsupervised machine learning approach for clustering is only adequate when there are many more samples available. I advise the authors to rely on the results from the hierarchical clustering for the purpose of identifying groups of samples.

Ln 210/260-261: What variables are the rows and columns of these tables?

Ln 217-219: Samples were assembled independently or was a cross-assembly performed?

Ln 236-238: This section needs to be expanded. Considering that a single scaffold might have multiple proteins, and these proteins might match multiple taxa, how exactly was taxonomic assignment performed? Was it simply the best hit (highest bitscore) among all proteins? There are tools to classify viral scaffolds with higher accuracy [9]⁠.

Ln 329-331: It is important to note that Siphoviruses are usually more abundant than Myoviruses in freshwater ecosystems while the opposite is true for marine ecosystems

Ln 635-638: You cannot claim this without extensively analysing the prevalence of the assembled viral sequences across multiple habitats at a global scale.

Several sentences of the results section belong in the discussion.

The link to the supplementary material did not work for me and therefore I was unable to assess this data and anything directly related to it

Figures 2 and 3: Consider using the log scale for counts or the row Z-score to better highlight differences among samples.

References:

1. Pons JC, Paez-Espino D, Riera G, Ivanova N, Kyrpides NC, Llabrés M. VPF-Class: Taxonomic assignment and host prediction of uncultivated viruses based on viral protein families. Valencia A, editor. Bioinformatics [Internet]. 2021;1–9. Available from: https://academic.oup.com/bioinformatics/advance-article/doi/10.1093/bioinformatics/btab026/6104829

2. Boeckaerts D, Stock M, Criel B, Gerstmans H, De Baets B, Briers Y. Predicting bacteriophage hosts based on sequences of annotated receptor-binding proteins. Sci Rep [Internet]. Nature Publishing Group UK; 2021;11:1–14. Available from: https://doi.org/10.1038/s41598-021-81063-4

3. Wang W, Ren J, Tang K, Dart E, Ignacio-Espinoza JC, Fuhrman JA, et al. A network-based integrated framework for predicting virus–prokaryote interactions. NAR Genomics Bioinforma [Internet]. Oxford University Press; 2020;2:505768. Available from: https://academic.oup.com/nargab/article/doi/10.1093/nargab/lqaa044/5861484

4. Edwards RA, McNair K, Faust K, Raes J, Dutilh BE. Computational approaches to predict bacteriophage–host relationships. Smith M, editor. FEMS Microbiol Rev [Internet]. 2016;40:258–72. Available from: https://academic.oup.com/femsre/article-lookup/doi/10.1093/femsre/fuv048

5. Galiez C, Siebert M, Enault F, Vincent J, Söding J. WIsH: who is the host? Predicting prokaryotic hosts from metagenomic phage contigs. Birol I, editor. Bioinformatics [Internet]. 2017;33:3113–4. Available from: https://academic.oup.com/bioinformatics/article-lookup/doi/10.1093/bioinformatics/btx383

6. Coutinho FH, Cabello-Yeves PJ, Gonzalez-Serrano R, Rosselli R, López-Pérez M, Zemskaya TI, et al. New viral biogeochemical roles revealed through metagenomic analysis of Lake Baikal. Microbiome [Internet]. Microbiome; 2020;8:163. Available from: https://microbiomejournal.biomedcentral.com/articles/10.1186/s40168-020-00936-4

7. Kieft K, Zhou Z, Anantharaman K. VIBRANT: Automated recovery, annotation and curation of microbial viruses, and evaluation of virome function from genomic sequences. bioRxiv. 2019;

8. Shaffer M, Borton MA, McGivern BB, Zayed AA, La Rosa SL, Solden LM, et al. DRAM for distilling microbial metabolism to automate the curation of microbiome function. Nucleic Acids Res. 2020;48:8883–900.

9. Bolduc B, Jang H Bin, Doulcier G, You Z-Q, Roux S, Sullivan MB. vConTACT: an iVirus tool to classify double-stranded DNA viruses that infect Archaea and Bacteria. PeerJ [Internet]. 2017;5:e3243. Available from: https://peerj.com/articles/3243

Author Response

Review 3

In this manuscript the authors evaluated the diversity of viruses at four sites from lake Baikal through metaviromics. This is a relevant topic with major knowledge gaps. Therefore, this work contributes to ongoing efforts to described the viral diversity of this ecosystem and the overall microbial diversity of freshwater habitats. The experiments are well designed and the results support the majority of conclusions. Nevertheless some issues must be addressed to make this manuscript suitable for publication

Major issues:

The viromes are derived from integrated samples from multiple depths. This is expected to significantly impact the results because the viral diversity is known to change with depth. Although this does not invalidate the results this limitation needs to be considered when interpreting the results and highlighted in the discussion.

  • The pros and cons of combining samples from different depths at each site were outlined in the Materials and Methods section and highlighted in Discussion (129-133, 718-720).

There is a lack of clarity in the methodology (detailed below) which make it particularly difficult to interpret the results.

Lack of relevant host information. No effort has been made to assign hosts to the assembled viral sequences. This is the most relevant question about the biology of these viruses and there are several tools available approaches achieve this in silico [1–5]⁠.

  • We carried out the host prediction analysis by two methods. One of them compared the taxonomic affiliation of scaffolds (as a particular virus) with information from the Virus-Host database (https://www.genome.jp/virushostdb/). Thus, we could identify the putative hosts among prokaryotes (archaea and bacteria) and eukaryotes (microalgae and protozoa). Another method used the VirHostMatcher-Net software from the list of software you recommend (publication [3] from your recommended list). In this case, we identified a broader spectrum of prokaryotic hosts. The results of these analyzeswere presented in the edited version of the manuscript (section 2.10 and 5; Lines 741-764).

Lack of relevant information about viral metabolic potential. Authors analysed the functional roles of viral proteins only superficially. Auxiliary metabolic genes, an important aspect of viral communities with some novel examples from lake Baikal [6]⁠, have not been analysed properly. This data is also fundamental for a proper understanding of viral contribution to ecosystem functioning and there are also tools designed to identify such genes in silico [7,8]⁠. Coupled with host prediction data, exploring the diversity of auxiliary metabolic genes would significantly enhance the relevance of this manuscript.

  • We performed an additional analysis to identify the auxiliary metabolic genes using the recommended VIBRANT program (publication [7] from your recommended list) (Lines 300-307). All results were added to the manuscript (Lines 477-536). The data obtained allowed us to make several interesting assumptions about the role of viruses in the functioning of the microbial community of Lake Baikal. Our suggestions were given in the Discussion section of the manuscript (Lines 782-814).

The section 2.11. “Comparison of viral scaffold taxonomic composition, predicted hosts, and AMGs” has been added to the Material and Methods. Some additional Figures (4, 5, 6A) and Supplementary Materials (Tables S8-S10, Figure S4) were also added.

Minor issues:

Ln 39-40: Not clear what is meant by wide and deep geography

  • This sentence was rewritten: “Marine studies encompass a wide variety of seas and oceans around the world, a large number of stations and samples from different depths, seasons, etc. [6]”.

Ln 43-45: Sentence too long, verbs and commas missing.

  • This sentence was divided: “Baikal is a unique, world's deepest, oldest, and largest by volume ancient oligotrophic freshwater lake. The lake characterized by an unusual climatic environment and the amazing biological diversity (mainly endemic flora and fauna) [19, 20]”.

Ln 69-70: Please elaborate: How exactly the environmental factors influence these viral communities?

  • It was specified: “the environmental factors specifying the trophic status of the lakes influence the diversity of viral communities and may be of greater importance than geographical patterns [24]”.

Ln 80 “viral shotgun metaviromes” is a redundancy, use “viral metagenomes” or “metaviromes”

  • It was taken into account.

Ln 161-164: Confusing sentence, plus the concept of virotype is not properly introduced.

  • This sentence was reworded: “The reads similar to the same viral genome (ID) from NCBI RefSeq database, were summarized and assigned to one virotype (virus species from the NCBI RefSeq database)”. (Lines 186-187).

Ln 152-174: This section is confusing and difficult to follow. It is unclear to me how the abundances based on genomic and proteomic data were combined. Also, there is no reference to this particular methodology being used before. It the authors want to introduce a new method it is important to describe it more clearly, justify its choice and demonstrate its superiority over more classical approaches of calculating viral abundances based on metagenomic data (such as percentage of mapped reads or RPKM)

  • We tried to rewrite this section of the manuscript and make it more understandable (section 2.5). Our taxonomic read identification algorithm works similar to the MetaVIR 2 service (https://bmcbioinformatics.biomedcentral.com/articles/10.1186/1471-2105-15-76). Only an additional improvement was made to compare the reads with the non-annotated genomes from the NCBI RefSeq database. This expanded the list of complete viral genomes involved in the analysis. The implemented approach to the identification of reads provided the use of the available computational resources with a large number of computational cores and a small amount of RAM (Lines 202-214).

Ln 192: Unclear what is meant by hidden diversity

  • Hidden diversity is potential number of viral species that could be detected by increasing the depth of reading. The corresponding comments were included in the text of the manuscript (Lines 222-224).

Ln 199-201: This type of unsupervised machine learning approach for clustering is only adequate when there are many more samples available. I advise the authors to rely on the results from the hierarchical clustering for the purpose of identifying groups of samples.

  • We added hierarchical clustering to highlight groups of samples. The changes were made to the Materials and Methods (Lines 237-239) and the Results sections (Lines 592-628) of the manuscript.

Ln 210/260-261: What variables (переменные) are the rows and columns of these tables?

  • The necessary clarifications (“Rows (virotypes/ protein functional categories) and columns (samples)”) were added to the manuscript (241-242, 297-298).

Ln 217-219: Samples were assembled independently or was a cross-assembly performed?

  • Cross-assembly of samples was performed, and clarification was made (Line 250).

Ln 236-238: This section needs to be expanded. Considering that a single scaffold might have multiple proteins, and these proteins might match multiple taxa, how exactly was taxonomic assignment performed? Was it simply the best hit (highest bitscore) among all proteins? There are tools to classify viral scaffolds with higher accuracy [9].

  • Bit score and other parameters (word size, gap open cost, gap extension cost, e-value, and identity) were used to identify scaffold proteins. Taxonomic assignment was based on the largest number of matching proteins or their similarity. This section was expanded and clarified (263-264). We also tried to use the vConTACT program from the publication [9] that you recommended, but we had problems with the links to the documentation on the website of this program, which did not work.

Ln 329-331: It is important to note that Siphoviruses are usually more abundant than Myoviruses in freshwater ecosystems while the opposite is true for marine ecosystems

  • It was specified (Lines 368-370).

Ln 635-638: You cannot claim this without extensively analysing the prevalence of the assembled viral sequences across multiple habitats at a global scale.

  • It was deleted.

Several sentences of the results section belong in the discussion.

  • The Results were checked, and some sentences and paragraphs were moved to the Discussion (Lines 645-699). 

The link to the supplementary material did not work for me and therefore I was unable to assess this data and anything directly related to it

  • We cannot explain this, but other Reviewers checked the Supplementary material (temporary link for Reviewers: https://figshare.com/s/68b7743e61cc33c620e3). If this problem persists, please let us know through the Editor, we will open the access.

Figures 2 and 3: Consider using the log scale for counts or the row Z-score to better highlight differences among samples.

  • The figures presented in the manuscript show the scale of the decimal logarithm for displaying color gradients. We experimented with the Z-score normalization, but the color display turned out to be very contrasting: in the blue-white-red color transition, most of the heat map field was displayed in white. We have now modified Figure 3 by making the color transition white-yellow-red on the log scale and added clarifications about the log scale to the captions (Lines 379-380, 498-499).

References:

  1. Pons JC, Paez-Espino D, Riera G, Ivanova N, Kyrpides NC, Llabrés M. VPF-Class: Taxonomic assignment and host prediction of uncultivated viruses based on viral protein families. Valencia A, editor. Bioinformatics [Internet]. 2021;1–9. Available from: https://academic.oup.com/bioinformatics/advance-article/doi/10.1093/bioinformatics/btab026/6104829
  2. Boeckaerts D, Stock M, Criel B, Gerstmans H, De Baets B, Briers Y. Predicting bacteriophage hosts based on sequences of annotated receptor-binding proteins. Sci Rep [Internet]. Nature Publishing Group UK; 2021;11:1–14. Available from: https://doi.org/10.1038/s41598-021-81063-4
  3. Wang W, Ren J, Tang K, Dart E, Ignacio-Espinoza JC, Fuhrman JA, et al. A network-based integrated framework for predicting virus–prokaryote interactions. NAR Genomics Bioinforma [Internet]. Oxford University Press; 2020;2:505768. Available from: https://academic.oup.com/nargab/article/doi/10.1093/nargab/lqaa044/5861484
  4. Edwards RA, McNair K, Faust K, Raes J, Dutilh BE. Computational approaches to predict bacteriophage–host relationships. Smith M, editor. FEMS Microbiol Rev [Internet]. 2016;40:258–72. Available from: https://academic.oup.com/femsre/article-lookup/doi/10.1093/femsre/fuv048
  5. Galiez C, Siebert M, Enault F, Vincent J, Söding J. WIsH: who is the host? Predicting prokaryotic hosts from metagenomic phage contigs. Birol I, editor. Bioinformatics [Internet]. 2017;33:3113–4. Available from: https://academic.oup.com/bioinformatics/article-lookup/doi/10.1093/bioinformatics/btx383
  6. Coutinho FH, Cabello-Yeves PJ, Gonzalez-Serrano R, Rosselli R, López-Pérez M, Zemskaya TI, et al. New viral biogeochemical roles revealed through metagenomic analysis of Lake Baikal. Microbiome [Internet]. Microbiome; 2020;8:163. Available from: https://microbiomejournal.biomedcentral.com/articles/10.1186/s40168-020-00936-4
  7. Kieft K, Zhou Z, Anantharaman K. VIBRANT: Automated recovery, annotation and curation of microbial viruses, and evaluation of virome function from genomic sequences. bioRxiv. 2019;
  8. Shaffer M, Borton MA, McGivern BB, Zayed AA, La Rosa SL, Solden LM, et al. DRAM for distilling microbial metabolism to automate the curation of microbiome function. Nucleic Acids Res. 2020;48:8883–900.
  9. Bolduc B, Jang H Bin, Doulcier G, You Z-Q, Roux S, Sullivan MB. vConTACT: an iVirus tool to classify double-stranded DNA viruses that infect Archaeaand Bacteria. PeerJ [Internet]. 2017;5:e3243. Available from: https://peerj.com/articles/3243

  • Thank you again for your detailed analysis, comments and objective criticism. The additional analysis of our data supplemented the manuscript with new interesting results and really increased the value and quality of our study.

Consequently, we used two tools that you recommended: VIBRANT (ref. 7) for viral contigs functional analysis and VirHostMatcher-Net (ref. 3) for virus-host prediction. Unfortunately, some other tools were poorly described, not worked or did not apply to our study. We would also like to add some comments to these programs you recommend and explain our choices or refusals concerning them.

[1] Pons JC, Paez-Espino D, Riera G, Ivanova N, Kyrpides NC, Llabrés M. VPF-Class: Taxonomic assignment and host prediction of uncultivated viruses based on viral protein families. Valencia A, editor. Bioinformatics [Internet]. 2021;1–9. 

-      The authors do not include the reference files in distribution https://github.com/biocom-uib/vpf-tools. Manual is very   short and it is unclear how to deal with "The most recent hmms file containing the HMMER models of VPFs (vpfsFile in data-index.yml) can be downloaded from IMG/VR". We will contact the authors for further study.

[2] Boeckaerts D, Stock M, Criel B, Gerstmans H, De Baets B, Briers Y. Predicting bacteriophage hosts based on sequences of annotated receptor-binding proteins. Sci Rep [Internet]. Nature Publishing Group UK; 2021;11:1–14.

-     This study mainly contains a description of the mathematical model. The authors tested this model on a small number of viral genomes, but this tool cannot predict the bacterial host for a diverse environmental viral metagenome.

[4] Edwards RA, McNair K, Faust K, Raes J, Dutilh BE. Computational approaches to predict bacteriophage–host relationships. Smith M, editor. FEMS Microbiol Rev [Internet]. 2016;40:258–72. 

-     This review paper was written five years ago. The review mainly deals with description of approaches and their comparison. VirHostMatcher-Net includes some of these approaches, and we decided not to use the tools cited in this review due to the lack of time.

[5] Galiez C, Siebert M, Enault F, Vincent J, Söding J. WIsH: who is the host? Predicting prokaryotic hosts from metagenomic phage contigs. Birol I, editor. Bioinformatics [Internet]. 2017;33:3113–4.

-      WiSH models and data are already included in the more powerful tool VirHostMatcher-Net, so we decided not to use WiSH separately.

Round 2

Reviewer 1 Report

The manuscript has been revised properly.